# Positive-Unlabeled Learning with Extreme Scarcity of Labeled Positives

**Yuanchao Dai** [1 2]   **Ximing Li** [1 2 3]   **Wei Wang** [3]   **Changchun Li** [1 2]   **Gang Niu** [3]   **Masashi Sugiyama** [3 4]

## Abstract

Positive-Unlabeled (PU) learning is a weakly-supervised paradigm that trains a binary classifier from labeled positive and unlabeled instances. In PU risk estimation, the empirical risk consists of an unlabeled term and a positive term. In this paper, we observe that when labeled positives are scarce, the risk deviation is dominated by the generalization bound of the positive term, which is composed of a complexity term governed by Rademacher complexity and a concentration term governed by the uniform range bound, leading to estimator instability. Based on this observation, we theoretically derive the sufficient sample threshold, defined as the smallest number of labeled positives required to achieve a target excess risk with high probability, and reveal its explicit dependence on both components. Inspired by this insight, we propose ScalePU, which incorporates variance regularization to induce a restricted sub-hypothesis space with reduced Rademacher complexity, and geometric regularization to encourage compact clustering of positive samples with a tighter effective range. Theoretical analysis demonstrates that both mechanisms effectively lower the threshold through improvements to different components of the bound. Experiments on eight benchmark datasets validate the effectiveness of ScalePU, with significant improvements under extreme label scarcity.

## 1. Introduction

Positive-Unlabeled (PU) learning is a weakly supervised binary classification paradigm, where the training data consist of labeled positive examples and unlabeled examples (du Plessis et al., 2015; Kiryo et al., 2017). This setting naturally arises in many real-world applications, where confirming positive instances is straightforward, yet obtaining verified negative labels proves to be impractical or prohibitively expensive, *e.g.,* medical diagnosis (Yang et al., 2012), text classification (Li & Liu, 2003), anomaly detection (Zhang et al., 2017b), and recommender systems (Hsieh et al., 2015).

Under the Selected Completely At Random (SCAR) assumption (Elkan & Noto, 2008), where labeled positive examples are drawn uniformly from the positive class distribution, the unbiased PU (uPU) risk estimator (du Plessis et al., 2015) demonstrated that the supervised classification risk can be reformulated using only positive and unlabeled data, leading to an unbiased risk estimator. Subsequent developments have largely followed two complementary directions: refining the risk estimation procedure to improve stability (Kiryo et al., 2017; Hammoudeh & Lowd, 2020; Zhao et al., 2022), and leveraging pseudo-labeling techniques to iteratively identify likely negative examples from the unlabeled set (Chen et al., 2020a; Li et al., 2022; Wang et al., 2023). Despite these advances, existing methods are predominantly designed and evaluated under settings where a sufficient number of labeled positive examples are available (du Plessis et al., 2015; Kiryo et al., 2017; Chen et al., 2020a; Li et al., 2022; Wang et al., 2023). However, when the number of labeled positive examples decreases, the positive term in the risk estimator becomes increasingly unreliable. According to McDiarmid's inequality (McDiarmid et al., 1989), the uniform deviation of the empirical risk is fundamentally governed by two factors: the Rademacher complexity of the hypothesis class and the uniform range bound of the positive term (Boucheron et al., 2003; 2013). In extremely label-scarce regimes, both factors become critical bottlenecks governing estimation stability.

This observation prompts a fundamental question: *what is the minimum number of labeled positive examples required for effective PU learning?* To formalize this question, we introduce the concept of the *sufficient sample threshold*, defined as the smallest number of labeled positive examples that guarantees achieving a target excess risk with high probability (Shalev-Shwartz & Ben-David, 2014). Our theoretical analysis reveals that this quantity depends on both the

[1]College of Computer Science and Technology, Jilin University, China [2]Key Laboratory of Symbolic Computation and Knowledge Engineering of Ministry of Education, Jilin University, China [3]RIKEN Center for Advanced Intelligence Project, Tokyo, Japan [4]The University of Tokyo, Japan. Correspondence to: Ximing Li <liximing86@gmail.com>.

*Proceedings of the 43rd International Conference on Machine Learning*, Seoul, South Korea. PMLR 306, 2026. Copyright 2026 by the author(s).

Rademacher complexity term and the concentration term involving the uniform range bound. This finding suggests that methods addressing both components have the potential to lower the sample complexity threshold, enabling learning with fewer labeled positive examples than conventional approaches require.

In this paper, building on this theoretical foundation, we propose ScalePU, a framework for PU learning under extreme label scarcity that targets both components of the generalization bound through complementary regularization mechanisms. Specifically, we first introduce variance regularization that constrains the functional variance over the positive distribution. By penalizing high variance, we effectively restrict the hypothesis space to a sub-class with reduced Rademacher complexity, thereby improving the first component of the bound. Second, recognizing that the concentration term is governed by the geometric dispersion in the representation space through the Lipschitz property, we incorporate geometric regularization to encourage positive examples to form a compact cluster. This geometric constraint directly reduces the effective range of the positive term, improving the second component of the bound.

Our main contributions can be summarized as follows.

- We theoretically characterize the instability of PU learning under label scarcity through both the Rademacher complexity and the uniform range bound of the positive term, and introduce the sufficient sample threshold as a metric for quantifying sample complexity.

- We propose variance regularization to reduce the Rademacher complexity and geometric regularization to reduce the effective range bound, jointly lowering the sufficient sample threshold.

- Extensive experiments on eight datasets demonstrate that our proposed method consistently outperforms state-of-the-art methods.

## 2. Related Work

In this section, we briefly review most related works on PU learning. Existing methods can be broadly categorized into two fundamental families based on how they handle unlabeled instances: disambiguation-free empirical risks and pseudo-labeling methods.

**Disambiguation-free Empirical Risks.** These methods approximate the expected risk of supervised learning without explicitly estimating labels for unlabeled instances. Under the Selected Completely At Random (SCAR) assumption, du Plessis et al. proposed an unbiased PU (uPU) risk estimator, which provides an unbiased estimator for the supervised

classification risk. However, uPU suffers from severe overfitting when using flexible models (Zhang et al., 2017a) due to potentially negative empirical risk terms. To address this limitation, the non-negative PU (nnPU) risk estimator (Kiryo et al., 2017) incorporates a non-negativity constraint using the max function, while the absolute-value PU (absPU) risk estimator (Hammoudeh & Lowd, 2020) applies an absolute-value constraint to prevent the risk estimates from becoming negative. Recently, the distribution alignment PU (Dist-PU) risk estimator (Zhao et al., 2022) reformulates the risk estimation from a label distribution perspective under symmetric losses, achieving improved stability.

**Pseudo-labeling Methods.** The basic idea of pseudo-labeling methods is estimating pseudo-labels for unlabeled instances and training the classifier in a self-training manner. Based on the pseudo-labeling techniques employed, these methods can be categorized into three groups. *Hard pseudo-labeling methods* assign discrete labels to unlabeled instances based on prediction confidence. Representative works include Rank Pruning, which iteratively identifies reliable negative examples from unlabeled data, and PU learning with negative sample selector (PULNS) (Luo et al., 2021), which designs effective negative sample selectors for hard label assignment. *Soft pseudo-labeling methods* (Chen et al., 2020a;b; Dai et al., 2025) maintain continuous confidence scores rather than hard assignments (Berthelot et al., 2019; Sohn et al., 2020; Berthelot et al., 2022; Zhang et al., 2021; Wang et al., 2022). Variational PU (VPU) learning (Chen et al., 2020a) introduces a variational framework that models pseudo-labels as latent variables, while self-boosted (Self-PU) learning (Chen et al., 2020b) employs evolving confidence scores throughout training with self-boosting mechanisms. *Data augmentation-enhanced methods* integrate pseudo-labeling with advanced augmentation techniques. PU learning with partially positive mixup ($P^3$Mix) (Li et al., 2022) identifies optimal mixup partners (Zhang et al., 2018; Verma et al., 2019) for PU learning, leveraging the interpolation principle to generate informative training samples. Recently, several sophisticated techniques have been developed to further improve pseudo-label quality, including holistic predictive trend exploitation (Wang et al., 2023), latent group-aware meta disambiguation (Long et al., 2024), and controlled probability boundary mechanisms (Li et al., 2024).

While these methods establish principled guiding ideas with theoretical guarantees (Bartlett & Mendelson, 2002; Ledoux & Talagrand, 2013), they do not explicitly address the instability that emerges when positive samples are scarce, which is the central focus of our work.

## 3. The Proposed Method

In this section, we introduce the proposed methods.

### 3.1. Preliminaries and Problem Setup

Let $\mathcal{X} \subseteq \mathbb{R}^d$ denote the input feature space and $\mathcal{Y} = \{-1, +1\}$ denote the binary label space. In the PU learning setting, we have access to a set of positive examples $\{x_i^{\mathrm{P}}\}_{i=1}^{n_{\mathrm{P}}} \overset{\text{i.i.d.}}{\sim} p(x \mid y = +1)$ and a set of unlabeled examples $\{x_j^{\mathrm{U}}\}_{j=1}^{n_{\mathrm{U}}} \overset{\text{i.i.d.}}{\sim} p(x)$ drawn from the marginal distribution. Let $\mathcal{X}_{\mathrm{P}} \subseteq \mathcal{X}$ denote the support of the positive class distribution $p(x \mid y = +1)$. We assume the class prior $\pi = p(y = +1)$ is known. Our goal is to learn a decision function $f : \mathcal{X} \to \mathbb{R}$ from a hypothesis class $\mathcal{H}$ that minimizes the expected classification risk $R(f) = \mathbb{E}_{(x,y) \sim p(x,y)}[\ell(y f(x))]$, where $\ell : \mathbb{R} \to [0, B]$ is a bounded, $L$-Lipschitz continuous surrogate loss function.

Under the Selected Completely At Random (SCAR) assumption (Elkan & Noto, 2008), the unbiased PU (uPU) empirical risk estimator (du Plessis et al., 2015) is formulated as

$$\widehat{R}_{\mathrm{PU}}(f) = \underbrace{\frac{1}{n_{\mathrm{U}}} \sum_{j=1}^{n_{\mathrm{U}}} \ell(-f(x_j^{\mathrm{U}}))}_{\text{unlabeled term}} + \underbrace{\frac{\pi}{n_{\mathrm{P}}} \sum_{i=1}^{n_{\mathrm{P}}} g_f(x_i^{\mathrm{P}})}_{\text{positive term}}, \quad (1)$$

where $g_f(x) = \ell(f(x)) - \ell(-f(x))$ captures the contribution of each positive example to the risk estimator. For notational convenience, we denote the empirical positive term as $\widehat{R}_{\mathrm{P}}(f) = \frac{\pi}{n_{\mathrm{P}}} \sum_{i=1}^{n_{\mathrm{P}}} g_f(x_i^{\mathrm{P}})$ with its expected risk as $R_{\mathrm{P}}(f) = \pi \mathbb{E}_{x \sim p(x|y=+1)}[g_f(x)]$.

### 3.2. The Source of Instability in Scarce PU Learning

We begin by analyzing the fundamental source of instability when learning from limited positive examples. In the label-scarce regime where $n_{\mathrm{P}} \ll n_{\mathrm{U}}$, the positive term $\widehat{R}_{\mathrm{P}}(f)$ dominates the estimation error.

Specifically, the generalization stability of this term can be characterized through McDiarmid's inequality (McDiarmid et al., 1989) combined with Rademacher complexity bounds (Bartlett & Mendelson, 2002). We consider the range of $g_f$ over the positive support, defined as $\mathrm{range}_{\mathrm{P}}(g_f) = \sup_{x \in \mathcal{X}_{\mathrm{P}}} g_f(x) - \inf_{x \in \mathcal{X}_{\mathrm{P}}} g_f(x)$, which satisfies the uniform bound $\mathrm{range}_{\mathrm{P}}(g_f) \leq 2B$ for all $f \in \mathcal{H}$. The uniform deviation between the empirical and expected positive risk satisfies, with probability at least $1 - \delta$,

$$\sup_{f \in \mathcal{H}} \left| \widehat{R}_{\mathrm{P}}(f) - R_{\mathrm{P}}(f) \right|$$

$$\leq \underbrace{4\pi L \mathfrak{R}_{n_{\mathrm{P}}}(\mathcal{H})}_{\text{complexity term}} + \underbrace{\pi B \sqrt{\frac{2 \ln(2/\delta)}{n_{\mathrm{P}}}}}_{\text{concentration term}}, \quad (2)$$

where $\mathfrak{R}_{n_{\mathrm{P}}}(\mathcal{H})$ denotes the Rademacher complexity of the hypothesis class. The complexity term quantifies the representational capacity of $\mathcal{H}$ over the positive distribution,

while the concentration term characterizes the estimation stability as restricted by the range bound $2B$. Both terms scale as $O(1/\sqrt{n_{\mathrm{P}}})$. When $n_{\mathrm{P}}$ is extremely small, their numerical magnitudes significantly outweigh the error of the unlabeled term, which scales as $O(1/\sqrt{n_{\mathrm{U}}})$. Consequently, even as the amount of unlabeled data approaches infinity, the scarcity of positive examples continues to induce severe risk fluctuations through Eq. (2).

**Definition 3.1.** Let $\mathcal{H} \subseteq \{f : \mathcal{X} \to \mathbb{R}\}$ denote a hypothesis class, and let $\ell : \mathbb{R} \to [0, B]$ be a surrogate loss function that is $L$-Lipschitz continuous and classification-calibrated for the 0-1 loss. Given a tolerance level $\epsilon > 0$ and a confidence parameter $\delta \in (0, 1)$, we define the sufficient sample threshold $P_{\epsilon, \delta; \mathrm{PU}}^*$ as the smallest positive example size $n_{\mathrm{P}}$ such that, when performing empirical risk minimization $\widehat{f} \in \arg\min_{f \in \mathcal{H}} \widehat{R}_{\mathrm{PU}}(f)$ using $n_{\mathrm{P}}$ positive examples drawn from $\mathcal{X}_{\mathrm{P}}$ and $n_{\mathrm{U}}$ unlabeled examples drawn from $\mathcal{X}$, the following probabilistic guarantee holds:

$$\Pr\left( R_{\mathrm{PU}}(\widehat{f}) - \inf_{f \in \mathcal{H}} R_{\mathrm{PU}}(f) \leq \epsilon \right) \geq 1 - \delta. \quad (3)$$

Intuitively, $P_{\epsilon, \delta; \mathrm{PU}}^*$ represents the critical threshold below which the estimation error introduced by insufficient positive labels dominates the learning process, leading to unstable risk estimation and degraded performance. At or above $P_{\epsilon, \delta; \mathrm{PU}}^*$, the uniform deviation over the hypothesis class is controlled with high probability, ensuring statistical consistency (Bartlett & Mendelson, 2002).

**Theorem 3.2.** *Under the setting of Definition 3.1, suppose the hypothesis class $\mathcal{H}$ has Rademacher complexity satisfying $\mathfrak{R}_n(\mathcal{H}) \leq \kappa/\sqrt{n}$ for some complexity parameter $\kappa \geq 0$ that depends on the structural properties of $\mathcal{H}$. Define the unlabeled-dependent constant $\Xi_{\mathrm{U}} = 4L\mathfrak{R}_{n_{\mathrm{U}}}(\mathcal{H}) + 2B\sqrt{\frac{2\ln(2/\delta)}{n_{\mathrm{U}}}}$. Provided that $\epsilon > \Xi_{\mathrm{U}}$, the sufficient sample threshold satisfies*

$$P_{\epsilon, \delta; \mathrm{PU}}^* = \left( \frac{8\pi L \kappa + 2\pi B\sqrt{2\ln(2/\delta)}}{\epsilon - \Xi_{\mathrm{U}}} \right)^2. \quad (4)$$

*Equivalently, for this bound to hold with probability at least $1 - \delta$, the number of positive examples satisfies*

$$n_{\mathrm{P}} \geq \left( \frac{8\pi L \kappa + 2\pi B\sqrt{2\ln(2/\delta)}}{\epsilon - \Xi_{\mathrm{U}}} \right)^2. \quad (5)$$

Theorem 3.2 reveals that the sample threshold $P_{\epsilon, \delta; \mathrm{PU}}^*$ depends on two distinct factors: the complexity parameter $\kappa$ through the Rademacher complexity term $8\pi L \kappa$, and the range bound $B$ through the concentration term $2\pi B\sqrt{2\ln(2/\delta)}$. This decomposition motivates our methods that we develop two complementary regularization mechanisms, each targeting one component of the bound.

## 3.3. PU Learning with Variance Regularization

The analysis in Theorem 3.2 demonstrates that the sufficient sample threshold is fundamentally governed by the Rademacher complexity term $8\pi L\kappa$. To lower this threshold, we propose to restrict the hypothesis space by leveraging the distributional properties of the positive samples. Recall that the Rademacher complexity $\mathfrak{R}_{n_{\mathrm{P}}}(\mathcal{G})$ quantifies the richness of the function class $\mathcal{G} = \{g_f : f \in \mathcal{H}\}$ by measuring its expected maximum correlation with random Rademacher sequences on the positive support. Crucially, this correlation capacity is inherently tied to the expected variation across the domain that if the functions in $\mathcal{G}$ exhibit minimal fluctuation on $\mathcal{X}_{\mathrm{P}}$, their ability to capture spurious correlations with random noise is fundamentally diminished, leading to a reduced Rademacher complexity.

This motivates a variance-constrained hypothesis class. Formally, it follows the Structural Risk Minimization (SRM) framework, inducing a sub-class $\mathcal{H}_\tau$ with restricted expected fluctuations to balance approximation and estimation errors (Cherkassky, 1997). Let $\mathrm{Var}_{\mathrm{P}}[g_f] = \mathbb{E}_{x \sim p(x|y=+1)}[(g_f(x) - \mathbb{E}[g_f])^2]$ denote the expected variance under the positive class distribution. We define the restricted class as

$$\mathcal{H}_\tau = \{f \in \mathcal{H} : \mathrm{Var}_{\mathrm{P}}[g_f] \leq \tau\} \subset \mathcal{H}. \quad (6)$$

By enforcing this variance constraint, we effectively prune high-complexity functions that exhibit excessive fluctuations over the positive distribution, thereby inducing a hypothesis space with reduced expressivity and improved generalization stability.

**Lemma 3.3.** *Let $\mathcal{G}_\tau = \{g_f : f \in \mathcal{H}_\tau\}$ denote the function class induced by the variance-constrained hypothesis class. The Rademacher complexity of $\mathcal{G}_\tau$ satisfies*

$$\mathfrak{R}_{n_{\mathrm{P}}}(\mathcal{G}_\tau) \leq \frac{\kappa_\tau}{\sqrt{n_{\mathrm{P}}}}, \quad (7)$$

*where $\kappa_\tau \leq \kappa$ is a complexity parameter that decreases as $\tau$ decreases.*

Since $\mathcal{H}_\tau \subseteq \mathcal{H}$, monotonicity of Rademacher complexity immediately gives $\kappa_\tau \leq \kappa$. The strict reduction $\kappa_\tau < \kappa$ for small $\tau$ can be established via Dudley's entropy integral (Dudley, 1967). Since variance-constrained functions satisfy $|g_f(x) - \mathbb{E}_{\mathrm{P}}[g_f]| \leq \sqrt{\tau}$ in an average sense, the effective integration range shrinks from $[0, B]$ to $[0, O(\sqrt{\tau})]$, yielding a reduced complexity parameter. The formal proof is provided in Appendix C.

While the constraint relies on the expected variance, we use the empirical variance $\widehat{\mathrm{Var}}_{\mathrm{P}}[g_f] = \frac{1}{n_{\mathrm{P}}} \sum_{i=1}^{n_{\mathrm{P}}} [g_f(x_i^{\mathrm{P}}) - \bar{g}]^2$ as a consistent proxy in practice, leading to the following optimization problem

$$\arg\min_{f \in \mathcal{H}} \widehat{R}_{\mathrm{PU}}(f) \quad \text{s.t.} \quad \widehat{\mathrm{Var}}_{\mathrm{P}}[g_f] \leq \tau, \quad (8)$$

where $\bar{g} = \frac{1}{n_{\mathrm{P}}} \sum_{i=1}^{n_{\mathrm{P}}} g_f(x_i^{\mathrm{P}})$.

By Lagrangian relaxation (Boyd & Vandenberghe, 2004), this constrained problem admits an equivalent Lagrangian form, which we refer to as PU learning with variance regularization (PUVR)

$$\arg\min_{f \in \mathcal{H}} \left\{ \widehat{R}_{\mathrm{PU}}(f) + \lambda \cdot \Omega_{\mathrm{var}}(f) \right\}, \quad (9)$$

where $\lambda > 0$ is the regularization strength and the variance regularization term $\Omega_{\mathrm{var}}(f)$ is defined as

$$\Omega_{\mathrm{var}}(f) = \pi \cdot \widehat{\mathrm{Var}}_{\mathrm{P}}[g_f]. \quad (10)$$

This term can be efficiently computed during training using automatic differentiation. In practice, we treat $\lambda$ as a tunable hyperparameter.

**Theorem 3.4.** *Under the setting of Definition 3.1, let $\widehat{f}_\tau$ be the solution to Eq.(9) with variance budget $\tau$. The solution $\widehat{f}_\tau$ lies in the restricted class $\mathcal{H}_\tau$ with Rademacher complexity parameter $\kappa_\tau \leq \kappa$. Provided that $\epsilon > \Xi_{\mathrm{U}}$, with probability at least $1 - \delta$, the sufficient sample threshold of PUVR satisfies*

$$P_{\epsilon,\delta;\mathrm{PUVR}}^*(\tau) = \left( \frac{8\pi L\kappa_\tau + 2\pi B\sqrt{2\ln(2/\delta)}}{\epsilon - \Xi_{\mathrm{U}}} \right)^2. \quad (11)$$

Compared to the standard PU learning bound in Theorem 3.2, PUVR achieves a reduced complexity parameter $\kappa_\tau \leq \kappa$ in the Rademacher complexity term. When $\kappa_\tau < \kappa$, we have $P_{\epsilon,\delta;\mathrm{PUVR}}^*(\tau) < P_{\epsilon,\delta;\mathrm{PU}}^*$. The reduction in sample complexity arises from restricting the hypothesis class to $\mathcal{H}_\tau$, where the variance constraint ensures reduced Rademacher complexity.

## 3.4. PU Learning with Geometric Regularization

While variance regularization reduces the Rademacher complexity term, the concentration term $2\pi B\sqrt{2\ln(2/\delta)}$ remains governed by the uniform range bound $2B$. We now introduce geometric regularization that directly reduces the effective range through representation compactness (Chapelle et al., 2006). Consider a feature mapping $h : \mathcal{X} \to \mathbb{R}^d$ and a classifier $f : \mathbb{R}^d \to \mathbb{R}$. We extend the definition of $g_f$ to the composed model as $g_{f,h}(x) = \ell(f(h(x))) - \ell(-f(h(x)))$. Since $\ell$ is $L$-Lipschitz and the classifier adopts a linear function $f(z) = w^\top z + b$ in the output layer, the composed function $g_{f,h}$ is $L_g$-Lipschitz continuous with respect to the feature representation, where $L_g = 2L\|w\|_2$. In practice, we apply weight decay regularization to the classifier weights, which implicitly bounds $\|w\|_2$ and thus controls $L_g$.

Under this setting, the range of $g_{f,h}$ over $\mathcal{X}_{\mathrm{P}}$ admits a geometric upper bound. Let $\mu_{\mathrm{P}} = \mathbb{E}_{x \sim p(x|y=+1)}[h(x)]$ denote

the expected positive class prototype in the representation space. For any two points $x, x' \in \mathcal{X}_P$, we have

$$
\begin{aligned}
|g_{f,h}(x) - g_{f,h}(x')| &\leq L_g \|h(x) - h(x')\| \\
&\leq L_g \left( \|h(x) - \mu_P\| + \|h(x') - \mu_P\| \right).
\end{aligned} \quad (12)
$$

This relationship reveals that the range of $g_{f,h}$ over $\mathcal{X}_P$ is governed by how tightly the positive distribution clusters around its expected prototype $\mu_P$ in the representation space. To formalize this, we consider a pre-defined geometrically constrained hypothesis class $\mathcal{H}_\rho^{\text{geo}}$ characterized by its expected compactness

$$
\mathcal{H}_\rho^{\text{geo}} = \left\{ (f, h) : \mathbb{E}_{x \sim p(x|y=+1)} \left[ \|h(x) - \mu_P\|^2 \right] \leq \rho^2 \right\}. \quad (13)
$$

For any $(f, h) \in \mathcal{H}_\rho^{\text{geo}}$, the effective range satisfies $\text{range}_P(g_{f,h}) \leq 2L_g\rho$, which replaces the uniform range bound $2B$ in the concentration analysis. The regularization term $\Omega_{\text{compact}}$, acting as an empirical surrogate for the constraint in $\mathcal{H}_\rho^{\text{geo}}$, serves as a structural constraint to ensure the learned solution remains within this restricted class

$$
\Omega_{\text{compact}}(h) = \frac{1}{n_P} \sum_{i=1}^{n_P} \|h(x_i^P) - c_P\|^2, \quad (14)
$$

where $c_P$ is employed as an empirical surrogate for $\mu_P$ in our optimization objective.

We further introduce an adaptive separation term preventing unlabeled examples from encroaching upon the positive cluster

$$
\Omega_{\text{sep}}(h) = \frac{1}{n_U} \sum_{j=1}^{n_U} w_j \cdot \max \left( 0, m - \|h(x_j^U) - c_P\|^2 \right), \quad (15)
$$

where $m > 0$ is a margin parameter and $w_j = 1 - s(h(x_j^U), c_P)$ is a soft-selection weight. The similarity function $s(z, c) = \exp(-\|z - c\|^2 / \sigma^2)$ with $s \in [0, 1]$ is an endogenous Gaussian kernel (Belkin et al., 2006) that enables $w_j$ to adaptively evolve alongside the learned representation $h$. This mechanism assigns lower weights to unlabeled examples with high similarity to the positive prototype, acknowledging their potential as unobserved positives.

By integrating these geometric regularizations with the variance-constrained objective, we propose the ScalePU, which minimizes the following joint objective

$$
\begin{aligned}
\min_{f,h} \quad & \widehat{R}_{\text{PU}}(f \circ h) + \lambda \cdot \Omega_{\text{var}}(f \circ h) \\
& + \gamma \cdot [\Omega_{\text{compact}}(h) + \beta \cdot \Omega_{\text{sep}}(h)], \quad (16)
\end{aligned}
$$

where $\lambda, \gamma, \beta > 0$ are hyperparameters controlling the respective regularization strengths. We note that standard

weight decay (Krogh & Hertz, 1991), which is applied to the classifier layer in our implementation, implicitly controls the Lipschitz constant $L_g = 2L\|w\|_2$, ensuring the geometric bound remains meaningful. The complete training procedure is summarized in Algorithm 1 in Appendix H.

**Theorem 3.5.** *Under the setting of Definition 3.1, let $\widehat{f} \circ \widehat{h}$ be the solution to the combined objective Eq.(16) with hyperparameters $(\lambda, \gamma, \beta)$. Suppose the solution achieves variance $Var_P[g_{f,h}] \leq \tau$ and compactness $\Omega_{compact}(h) \leq \rho^2$. Further assume the classifier $f(z) = w^\top z + b$ satisfies $\|w\|_2 \leq W$ (ensured by weight decay), so that $g_{f,h}$ is $L_g$-Lipschitz continuous in the feature space with $L_g = 2LW$. The solution lies in the intersection $\mathcal{H}_\tau \cap \mathcal{H}_\rho^{geo}$, with reduced complexity parameter $\kappa_\tau$ and effective range $2L_g\rho$. Provided that $\epsilon > \Xi_U$, the sufficient sample threshold satisfies*

$$
P_{\epsilon,\delta;\text{ScalePU}}^*(\tau, \rho) = \left( \frac{8\pi L \kappa_\tau + 2\pi L_g \rho \sqrt{2\ln(2/\delta)}}{\epsilon - \Xi_U} \right)^2. \quad (17)
$$

*When $\kappa_\tau < \kappa$ and $L_g\rho < B$, we have $P_{\epsilon,\delta;\text{ScalePU}}^*(\lambda, \gamma, \beta) < P_{\epsilon,\delta;\text{PU}}^*$.*

Theorem 3.5 formally substantiates the synergistic effect of our dual regularization framework in mitigating the instability of PU learning under extreme scarcity. Specifically, the generalization bound is refined through two complementary pathways that variance regularization restricts the hypothesis space to a subset with reduced Rademacher complexity ($\kappa \to \kappa_\tau$), while geometric regularization leverages the Lipschitz property to compress the effective range ($B \to L_g\rho$). This joint mechanism simultaneously addresses the structural complexity and the distributional concentration of the positive risk estimator. By yielding a strictly lower sufficient sample threshold, ScalePU ensures statistical consistency in extreme regimes.

## 4. Experiments

In this section, we conduct comprehensive experiments to evaluate the effectiveness of our proposed method.

### 4.1. Experimental Settings

**Datasets.** We use eight diverse datasets covering both standard benchmarks and domain-specific applications. Specifically, we employ four widely-used image datasets: **CIFAR-10** (Krizhevsky & Hinton, 2009), **Fashion-MNIST** (F-MNIST) (Xiao et al., 2017), **STL-10** (Coates et al., 2011), and **ImageNette** (Deng et al., 2009), along with the **Alzheimer** MRI dataset. To create binary PU learning scenarios from multi-class datasets, we partition the original categories into positive and negative groups, following established conventions in the literature (Kiryo et al.,

*Table 1.* Summary of PU datasets. We report input dimensionality, dataset splits, class prior $\pi$, positive class composition, and the neural network architecture employed for each dataset.

| Dataset | Dimension | Train/Test | $\pi$ | Positive Classes | Architecture |
|---|---|---|---|---|---|
| CIFAR-10-1 | $3 \times 32 \times 32$ | 50,000 / 10,000 | 0.4 | $\{0, 1, 8, 9\}$ | CNN-7 (CIFAR) |
| CIFAR-10-2 | $3 \times 32 \times 32$ | 50,000 / 10,000 | 0.6 | $\{2, 3, 4, 5, 6, 7\}$ | CNN-7 (CIFAR) |
| F-MNIST-1 | $28 \times 28$ | 60,000 / 10,000 | 0.4 | $\{0, 2, 4, 6\}$ | LeNet-5 |
| F-MNIST-2 | $28 \times 28$ | 60,000 / 10,000 | 0.6 | $\{1, 3, 5, 7, 8, 9\}$ | LeNet-5 |
| STL-10-1 | $3 \times 96 \times 96$ | 105,000 / 8,000 | 0.506 | $\{0, 2, 3, 8, 9\}$ | CNN-7 (STL) |
| STL-10-2 | $3 \times 96 \times 96$ | 105,000 / 8,000 | 0.494 | $\{1, 4, 5, 6, 7\}$ | CNN-7 (STL) |
| ImageNette | $3 \times 224 \times 224$ | 9,469 / 3,925 | 0.5 | $\{0, 1, 2, 8, 9\}$ | ResNet-50 |
| Alzheimer | $3 \times 224 \times 224$ | 5,121 / 1,279 | 0.5 | $\{0, 1, 3\}$ | ResNet-50 |

2017; Wang et al., 2023). The comprehensive summary of PU datasets and detailed model settings for ScalePU are provided in Appendix E.

To assess performance under extreme label scarcity, we vary the number of labeled positive samples as $n_P \in \{10, 20, 50, 100\}$ while keeping the unlabeled set fixed. All experiments are conducted using 5-fold cross-validation, and we report mean accuracy with standard deviation.

**Baselines.** We compare ScalePU against nine representative PU learning methods from two paradigms: disambiguation-free risk estimators including uPU (du Plessis et al., 2015), nnPU (Kiryo et al., 2017), abs-PU (Hammoudeh & Lowd, 2020), and DC-PU (Li et al., 2025); pseudo-labeling methods including VPU (Chen et al., 2020a), P³Mix (Li et al., 2022), HolisticPU (Wang et al., 2023), LaGAM (Long et al., 2024), and PUL-CPBF (Li et al., 2024). Additionally, we include PUVR as an ablation baseline, which represents our method with only variance regularization (Eq. 9) but without geometric regularization. Detailed descriptions of all baseline methods are provided in Appendix F.

### 4.2. Main Results

Table 2 presents classification accuracy across three datasets (*i.e.,* CIFAR-10, F-MNIST, and STL-10) and sample size configurations. ScalePU achieves the best performance in most experimental settings, with particularly pronounced advantages when positive samples are extremely scarce ($n_P \leq 20$). For example, on CIFAR-10-1 with only 10 positive samples, ScalePU reaches 67.5% accuracy, outperforming the second-best method by a clear margin and exceeding nnPU by 3.6%. Moreover, the performance gap between ScalePU and other methods becomes larger as positive samples become scarcer. This observation matches our theoretical analysis: when $n_P$ is small, the generalization bound becomes loose, and methods without explicit complexity and range control suffer more severely. As $n_P$ increases to 100, the gap narrows since both terms in the bound become tighter.

It is worth noting that PUVR, which contains only variance

regularization without geometric constraints, already outperforms most baselines. This validates our key insight that controlling the Rademacher complexity through variance constraints is essential for PU learning with limited positive data. ScalePU further improves PUVR by adding geometric regularization, which reduces the effective range through representation compactness (He et al., 2016). The only exception to ScalePU's dominance occurs in a few cases where LaGAM achieves comparable results, particularly when $n_P = 100$ and the variance issue is less severe.

In contrast, pseudo-labeling methods such as VPU and P³Mix show unstable behavior under extreme label scarcity. These approaches rely on confident model predictions to generate pseudo-labels for self-training, which becomes unreliable when there are too few positive examples to learn meaningful patterns. As a result, their performance drops sharply at $n_P = 10$, sometimes falling close to random guessing. This highlights the advantage of our approach, which directly addresses the statistical challenge rather than depending on prediction quality.

**Large-Scale Evaluation.** Table 3 reports results on ImageNette and Alzheimer, two high-resolution datasets that present additional challenges due to increased image complexity and domain-specific characteristics. On ImageNette, ScalePU demonstrates substantial improvements over all baselines across all sample sizes. With $n_P = 100$, ScalePU achieves 59.8% accuracy, representing a 4.7% absolute improvement over the second-best method (LaGAM at 55.1%). This gap widens as label scarcity increases: at $n_P = 10$, ScalePU attains 54.1% compared to LaGAM's 53.2%, while pseudo-labeling methods VPU and P³Mix collapse to near-random performance (around 51-52%). The Alzheimer dataset exhibits similar trends, where ScalePU consistently achieves the highest accuracy across all configurations. Notably, most baseline methods struggle on this medical imaging task, with VPU, P³Mix, and PUL-CPBF failing entirely (accuracy is approximately 50%). This failure can be attributed to the inherent difficulty of generating reliable pseudo-labels in specialized domains where pre-trained representations transfer poorly. In contrast, ScalePU maintains stable performance even at $n_P = 10$ (54.8%), demonstrat-

*Table 2.* Results of classification accuracy (mean±std) on six benchmark PU datasets. The highest scores are shown in bold.

| Dataset | Method | CIFAR-10-1 | CIFAR-10-2 | F-MNIST-1 | F-MNIST-2 | STL-10-1 | STL-10-2 |
|---|---|---|---|---|---|---|---|
| 10 | uPU | 0.606±0.002 | 0.406±0.005 | 0.672±0.054 | 0.413±0.011 | 0.606±0.045 | 0.533±0.055 |
| | nnPU | 0.639±0.014 | 0.577±0.045 | 0.712±0.072 | 0.503±0.082 | 0.609±0.052 | 0.555±0.064 |
| | abs-PU | 0.652±0.033 | 0.582±0.044 | 0.756±0.049 | 0.555±0.089 | 0.603±0.056 | 0.563±0.069 |
| | VPU | 0.600±0.000 | 0.400±0.000 | 0.600±0.000 | 0.400±0.000 | 0.529±0.035 | 0.508±0.008 |
| | P³MIX | 0.601±0.001 | 0.404±0.004 | 0.686±0.077 | 0.553±0.116 | 0.504±0.005 | 0.502±0.001 |
| | HolisticPU | 0.602±0.001 | 0.403±0.001 | 0.662±0.053 | 0.587±0.064 | 0.512±0.010 | 0.502±0.001 |
| | LaGAM | 0.673±0.011 | 0.484±0.031 | 0.751±0.022 | 0.575±0.037 | 0.605±0.062 | 0.583±0.056 |
| | PUL-CPBF | 0.600±0.000 | 0.560±0.080 | 0.686±0.064 | **0.585±0.064** | 0.500±0.000 | 0.500±0.000 |
| | DC-PU | 0.640±0.015 | 0.584±0.018 | 0.713±0.070 | 0.552±0.041 | 0.551±0.029 | 0.534±0.035 |
| | PUVR | 0.654±0.032 | 0.585±0.041 | 0.758±0.049 | 0.557±0.089 | **0.610±0.059** | 0.572±0.079 |
| | ScalePU | **0.675±0.030** | **0.593±0.034** | **0.775±0.039** | 0.583±0.050 | **0.610±0.059** | **0.584±0.043** |
| 20 | uPU | 0.617±0.011 | 0.437±0.026 | 0.680±0.055 | 0.427±0.020 | 0.596±0.064 | 0.544±0.037 |
| | nnPU | 0.710±0.033 | 0.674±0.051 | 0.805±0.048 | 0.621±0.097 | 0.610±0.080 | 0.589±0.050 |
| | abs-PU | 0.711±0.038 | 0.676±0.054 | 0.844±0.032 | 0.696±0.075 | 0.601±0.083 | 0.598±0.049 |
| | VPU | 0.604±0.004 | 0.401±0.001 | 0.600±0.000 | 0.400±0.000 | 0.547±0.033 | 0.512±0.009 |
| | P³MIX | 0.612±0.005 | 0.426±0.030 | 0.836±0.044 | 0.714±0.106 | 0.527±0.019 | 0.525±0.025 |
| | HolisticPU | 0.602±0.001 | 0.402±0.002 | 0.778±0.078 | 0.562±0.078 | 0.505±0.002 | 0.500±0.000 |
| | LaGAM | 0.675±0.016 | 0.499±0.030 | 0.799±0.022 | 0.597±0.044 | 0.601±0.052 | 0.557±0.031 |
| | PUL-CPBF | 0.600±0.000 | 0.560±0.080 | 0.784±0.088 | 0.730±0.056 | 0.545±0.057 | 0.500±0.000 |
| | DC-PU | 0.662±0.057 | 0.648±0.059 | 0.806±0.048 | 0.670±0.059 | 0.604±0.047 | 0.593±0.030 |
| | PUVR | 0.713±0.037 | 0.706±0.040 | 0.858±0.015 | 0.721±0.062 | 0.609±0.074 | 0.607±0.044 |
| | ScalePU | **0.716±0.038** | **0.712±0.048** | **0.865±0.016** | **0.742±0.045** | **0.616±0.062** | **0.611±0.042** |
| 50 | uPU | 0.661±0.032 | 0.596±0.065 | 0.836±0.059 | 0.607±0.069 | 0.663±0.049 | 0.520±0.016 |
| | nnPU | 0.737±0.039 | 0.711±0.054 | 0.888±0.032 | 0.768±0.106 | 0.676±0.049 | 0.556±0.075 |
| | abs-PU | 0.744±0.029 | 0.713±0.054 | 0.899±0.023 | 0.847±0.046 | 0.678±0.050 | 0.583±0.075 |
| | VPU | 0.620±0.016 | 0.404±0.001 | 0.601±0.000 | 0.401±0.001 | 0.629±0.027 | 0.584±0.062 |
| | P³MIX | 0.655±0.030 | 0.503±0.059 | 0.898±0.025 | **0.888±0.032** | 0.636±0.036 | 0.615±0.015 |
| | HolisticPU | 0.606±0.003 | 0.406±0.002 | 0.811±0.026 | 0.778±0.026 | 0.519±0.011 | 0.502±0.002 |
| | LaGAM | 0.662±0.013 | 0.514±0.008 | 0.819±0.022 | 0.749±0.011 | 0.587±0.024 | 0.566±0.009 |
| | PUL-CPBF | 0.600±0.000 | 0.560±0.080 | 0.824±0.081 | 0.846±0.064 | 0.663±0.056 | 0.500±0.000 |
| | DC-PU | 0.720±0.072 | 0.712±0.052 | 0.889±0.031 | 0.780±0.129 | 0.681±0.022 | 0.600±0.083 |
| | PUVR | 0.741±0.035 | 0.713±0.052 | 0.900±0.023 | 0.849±0.044 | 0.681±0.046 | 0.606±0.061 |
| | ScalePU | **0.754±0.023** | **0.719±0.068** | **0.901±0.022** | 0.857±0.037 | **0.684±0.042** | **0.647±0.060** |
| 100 | uPU | 0.700±0.014 | 0.645±0.051 | 0.855±0.068 | 0.606±0.080 | 0.701±0.037 | 0.636±0.058 |
| | nnPU | 0.764±0.026 | 0.766±0.024 | 0.922±0.010 | 0.880±0.041 | 0.705±0.042 | 0.679±0.065 |
| | abs-PU | 0.763±0.029 | 0.768±0.024 | 0.926±0.008 | 0.906±0.017 | 0.704±0.044 | 0.679±0.061 |
| | VPU | 0.722±0.030 | 0.514±0.051 | 0.613±0.009 | 0.404±0.002 | 0.692±0.021 | 0.655±0.056 |
| | P³MIX | 0.763±0.036 | 0.696±0.041 | 0.918±0.017 | **0.916±0.018** | 0.695±0.029 | **0.720±0.041** |
| | HolisticPU | 0.652±0.011 | 0.412±0.004 | 0.846±0.024 | 0.766±0.053 | 0.528±0.010 | 0.504±0.001 |
| | LaGAM | 0.689±0.008 | 0.518±0.020 | 0.896±0.007 | 0.749±0.011 | 0.613±0.018 | 0.586±0.037 |
| | PUL-CPBF | 0.600±0.000 | 0.600±0.000 | 0.890±0.037 | 0.853±0.046 | 0.709±0.043 | 0.500±0.000 |
| | DC-PU | 0.739±0.012 | 0.750±0.018 | 0.922±0.010 | 0.881±0.041 | 0.702±0.031 | 0.675±0.037 |
| | PUVR | 0.764±0.030 | 0.770±0.024 | 0.927±0.008 | 0.907±0.017 | 0.711±0.044 | 0.685±0.063 |
| | ScalePU | **0.789±0.008** | **0.777±0.020** | **0.929±0.005** | 0.907±0.017 | **0.713±0.038** | 0.705±0.038 |

ing its robustness in challenging real-world scenarios.

### 4.3. Ablation Study

To validate the contribution of different components, we conduct an ablation study on eight PU datasets. Table 4 presents the results for three variants, where we evaluate two key components: (a) variance regularization (Var. Reg.) and (b) geometric regularization (Geo. Reg.).

The results demonstrate that each component contributes to the overall performance. Using variance regularization alone yields competitive results on datasets such as F-MNIST-2 and Alzheimer, while geometric regularization alone shows stronger performance on high-resolution datasets including ImageNette, improving accuracy from 0.546 to 0.591 at $n_P = 100$. Combining both regularization terms consistently achieves the best results across all

*Table 3.* Results of classification accuracy (mean±std) on two large-scale PU datasets (*i.e.,* Alzheimer and ImageNette). The highest scores are indicated in bold.

| Dataset | Method | 100 | 50 | 20 | 10 |
|---|---|---|---|---|---|
| Alzheimer | uPU | 0.515±0.014 | 0.505±0.004 | 0.506±0.006 | 0.504±0.009 |
| | nnPU | 0.533±0.036 | 0.553±0.073 | 0.506±0.009 | 0.513±0.062 |
| | abs-PU | 0.566±0.019 | 0.568±0.048 | 0.545±0.037 | 0.513±0.052 |
| | VPU | 0.501±0.000 | 0.500±0.000 | 0.500±0.000 | 0.500±0.000 |
| | P3MIX | 0.501±0.000 | 0.500±0.000 | 0.500±0.000 | 0.500±0.000 |
| | HolisticPU | 0.503±0.012 | 0.497±0.012 | 0.506±0.031 | 0.497±0.007 |
| | LaGAM | 0.555±0.007 | 0.539±0.002 | 0.541±0.010 | 0.521±0.004 |
| | PUL-CPBF | 0.500±0.000 | 0.500±0.000 | 0.500±0.000 | 0.500±0.000 |
| | DC-PU | 0.580±0.027 | 0.570±0.040 | 0.537±0.039 | 0.547±0.055 |
| | PUVR | 0.574±0.030 | **0.579±0.026** | 0.566±0.029 | 0.533±0.031 |
| | ScalePU | **0.583±0.021** | **0.579±0.026** | **0.569±0.030** | **0.548±0.058** |
| ImageNette | uPU | 0.511±0.002 | 0.521±0.007 | 0.509±0.000 | 0.511±0.002 |
| | nnPU | 0.534±0.021 | 0.535±0.014 | 0.521±0.008 | 0.508±0.019 |
| | abs-PU | 0.537±0.016 | 0.536±0.013 | 0.521±0.008 | 0.525±0.015 |
| | VPU | 0.524±0.004 | 0.529±0.010 | 0.511±0.001 | 0.509±0.000 |
| | P3MIX | 0.525±0.057 | 0.520±0.033 | 0.546±0.043 | 0.520±0.028 |
| | HolisticPU | 0.512±0.001 | 0.511±0.002 | 0.512±0.002 | 0.515±0.006 |
| | LaGAM | 0.551±0.011 | 0.549±0.006 | 0.553±0.016 | 0.532±0.032 |
| | PUL-CPBF | 0.509±0.000 | 0.509±0.000 | 0.509±0.000 | 0.509±0.001 |
| | DC-PU | 0.544±0.013 | 0.533±0.017 | 0.524±0.013 | 0.520±0.009 |
| | PUVR | 0.546±0.018 | 0.541±0.023 | 0.528±0.013 | 0.525±0.019 |
| | ScalePU | **0.598±0.008** | **0.569±0.028** | **0.560±0.049** | **0.541±0.018** |

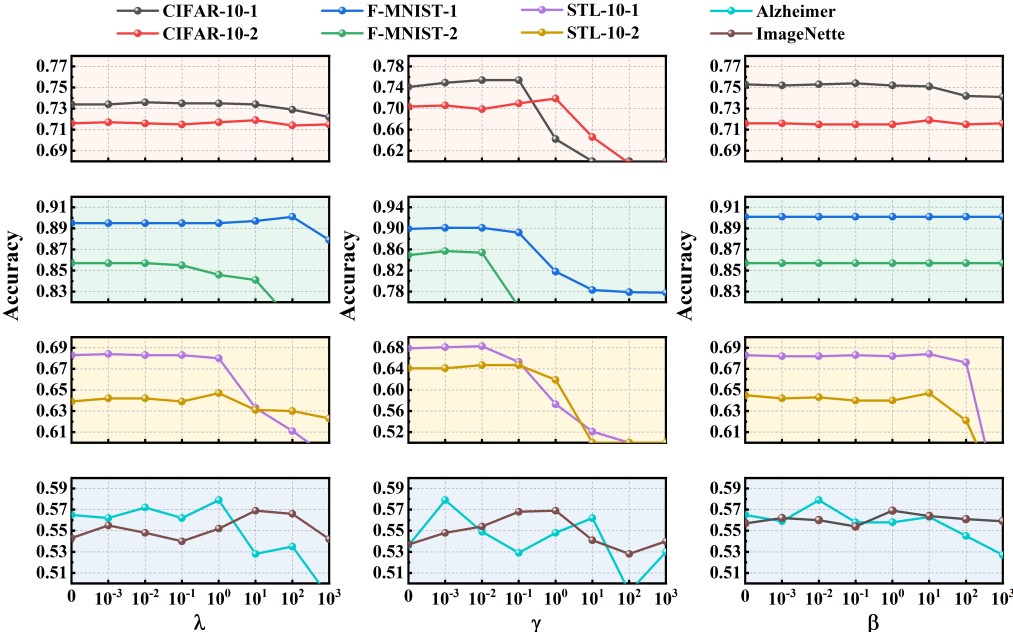

*Figure 1.* Parameter sensitivity analysis of parameters $\{\lambda, \gamma, \beta\}$ on eight PU datasets when $n_P = 50$.

configurations, with improvements most pronounced under extreme label scarcity. These results validate the complementary nature of the two mechanisms. Figure 1 further supplements the ablation analysis by showing the results when each regularization parameter is set to zero.

## 4.4. Parameter Sensitivity

Figure 1 presents the sensitivity analysis of the three key hyperparameters $\{\lambda, \gamma, \beta\}$ across all eight PU datasets, where each parameter varies from $10^{-3}$ to $10^3$.

*Table 4.* Experimental results of ablation study on eight PU datasets, where the best performance is shown in bold.

| Dataset | Var. Reg. | Geo. Reg. | 100 | 50 | 20 | 10 |
|---|---|---|---|---|---|---|
| CIFAR-10-1 | ✓ | | 0.764±0.030 | 0.741±0.035 | 0.713±0.037 | 0.654±0.032 |
| | | ✓ | **0.789±0.009** | **0.754±0.023** | 0.715±0.044 | 0.674±0.029 |
| | ✓ | ✓ | **0.789±0.008** | **0.754±0.023** | **0.716±0.038** | **0.675±0.030** |
| CIFAR-10-2 | ✓ | | 0.770±0.024 | 0.713±0.052 | 0.706±0.040 | 0.585±0.041 |
| | | ✓ | 0.773±0.021 | 0.703±0.046 | 0.680±0.057 | 0.589±0.027 |
| | ✓ | ✓ | **0.777±0.020** | **0.719±0.068** | **0.712±0.048** | **0.593±0.034** |
| F-MNIST-1 | ✓ | | 0.927±0.008 | 0.900±0.023 | 0.858±0.015 | 0.758±0.049 |
| | | ✓ | 0.926±0.008 | **0.901±0.022** | **0.868±0.024** | **0.775±0.038** |
| | ✓ | ✓ | **0.929±0.005** | **0.901±0.022** | 0.865±0.016 | **0.775±0.039** |
| F-MNIST-2 | ✓ | | **0.907±0.017** | 0.849±0.044 | 0.721±0.062 | 0.557±0.089 |
| | | ✓ | 0.902±0.016 | 0.851±0.037 | 0.729±0.039 | 0.565±0.095 |
| | ✓ | ✓ | **0.907±0.017** | **0.857±0.037** | **0.742±0.045** | **0.583±0.050** |
| STL-10-1 | ✓ | | 0.711±0.044 | 0.681±0.046 | 0.609±0.074 | 0.610±0.059 |
| | | ✓ | 0.712±0.040 | 0.676±0.057 | 0.613±0.070 | 0.605±0.056 |
| | ✓ | ✓ | **0.713±0.038** | **0.684±0.042** | **0.616±0.062** | **0.610±0.059** |
| STL-10-2 | ✓ | | 0.685±0.063 | 0.606±0.061 | 0.607±0.044 | 0.572±0.079 |
| | | ✓ | 0.702±0.043 | 0.625±0.085 | 0.610±0.041 | 0.581±0.038 |
| | ✓ | ✓ | **0.705±0.038** | **0.647±0.060** | **0.611±0.042** | **0.584±0.043** |
| Alzheimer | ✓ | | 0.574±0.030 | 0.579±0.026 | 0.566±0.029 | 0.533±0.031 |
| | | ✓ | 0.578±0.023 | 0.557±0.027 | 0.531±0.009 | 0.525±0.013 |
| | ✓ | ✓ | **0.583±0.021** | **0.579±0.026** | **0.569±0.030** | **0.548±0.058** |
| ImageNette | ✓ | | 0.546±0.018 | 0.541±0.023 | 0.528±0.013 | 0.525±0.019 |
| | | ✓ | 0.591±0.024 | 0.557±0.041 | 0.547±0.039 | 0.538±0.065 |
| | ✓ | ✓ | **0.598±0.008** | **0.569±0.028** | **0.560±0.049** | **0.541±0.018** |

For the variance regularization coefficient $\lambda$, performance generally improves as $\lambda$ increases from zero and stabilizes within the range $\lambda \in [10^{-1}, 10^1]$. Excessively large values $(\lambda > 10^2)$ lead to performance degradation due to over-regularization. The geometric regularization strength $\gamma$ shows dataset-dependent behavior, with performance remaining stable across $\gamma \in [10^{-2}, 10^1]$ on most datasets. Large-scale datasets (ImageNette and Alzheimer) benefit more from geometric regularization. The separation weight $\beta$ exhibits the most stable behavior, with performance curves remaining relatively flat for $\beta \in [10^{-1}, 10^1]$ across all datasets. Based on these observations, we recommend $\lambda \in [10^{-1}, 10^0]$, $\gamma \in [10^{-2}, 10^0]$, and $\beta \in [10^{-1}, 10^1]$ as reasonable default ranges.

## 5. Conclusion

This paper addressed the challenge of PU learning when labeled positive examples are extremely scarce. We identified that the generalization bound consists of a complexity term and a concentration term, both of which govern estimation stability under limited supervision, and formalized this through the sufficient sample threshold. To reduce this threshold, we developed ScalePU, which combines variance regularization to induce a restricted hypothesis class with reduced Rademacher complexity, and geometric regulariza-

tion to reduce the effective range through representation compactness. Extensive experiments across eight datasets confirmed that ScalePU achieves substantial improvements over existing methods.

## Acknowledgements

This work was supported in part by the National Natural Science Foundation of China (No.62276113), the JST ASPIRE Grant Number JPMJAP2405, and the Scientific and Technological Research Project of the Department of Education of Jilin Province (No.JJKH20262229BS).

## Impact Statement

The ScalePU framework proposed in this paper significantly enhances model stability and reliability in scenarios of extreme label scarcity through dual variance and geometric regularization. By theoretically characterizing the sources of instability in risk estimation, this research demonstrates substantial potential for practical applications in fields where negative samples are difficult to obtain, such as medical imaging and fraud detection. However, the model still largely relies on the SCAR sampling assumption, which may affect the fairness of the results if the original data contains biases. Moreover, in real-world deployment, users should

acknowledge the algorithm's role as a supportive component in decision-making to maintain individual autonomy and prevent excessive technical dependence. Ultimately, ScalePU provides a robust tool for weakly supervised learning and encourages a more stable approach to automated decision-making in critical domains.

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

# A. Proof of Theorem 3.2

We establish uniform convergence bounds for the PU risk estimator over the hypothesis class $\mathcal{H}$ using McDiarmid's inequality. Let $\mathcal{X}_P \subseteq \mathcal{X}$ denote the support of the positive class distribution $p(x \mid y = +1)$. For any $f \in \mathcal{H}$, the deviation between the empirical and true PU risks can be decomposed according to the structure of the PU risk as

$$
\begin{aligned}
\widehat{R}_{\mathrm{PU}}(f) - R_{\mathrm{PU}}(f) &= \left[ \pi \widehat{\mathbb{E}}_P[\ell(f(x^P))] + \widehat{\mathbb{E}}_U[\ell(-f(x^U))] - \pi \widehat{\mathbb{E}}_P[\ell(-f(x^P))] \right] \\
&\quad - \left[ \pi \mathbb{E}_P[\ell(f(x^P))] + \mathbb{E}_U[\ell(-f(x^U))] - \pi \mathbb{E}_P[\ell(-f(x^P))] \right] \\
&= \underbrace{\left( \widehat{\mathbb{E}}_U - \mathbb{E}_U \right) [\ell(-f(x^U))]}_{\text{unlabeled term}} + \pi \underbrace{\left( \widehat{\mathbb{E}}_P - \mathbb{E}_P \right) [g_f(x^P)]}_{\text{positive term}},
\end{aligned}
\tag{18}
$$

where $g_f(x) = \ell(f(x)) - \ell(-f(x))$ is defined as in the main text. Since $\ell : \mathbb{R} \to [0, B]$ is bounded, the function $g_f$ satisfies $g_f(x) \in [-B, B]$ for all $x \in \mathcal{X}$. Recall from the main text that the range of $g_f$ over the positive support is defined as $\mathrm{range}_P(g_f) = \sup_{x \in \mathcal{X}_P} g_f(x) - \inf_{x \in \mathcal{X}_P} g_f(x)$. Since $g_f(x) \in [-B, B]$, we obtain the uniform range bound $\mathrm{range}_P(g_f) \le 2B$ for all $f \in \mathcal{H}$.

We apply McDiarmid's inequality to bound the uniform deviation over $\mathcal{H}$. Define

$$
\Phi = \sup_{f \in \mathcal{H}} \left| \widehat{R}_{\mathrm{PU}}(f) - R_{\mathrm{PU}}(f) \right|
\tag{19}
$$

as a function of the combined sample $(\{x_i^U\}_{i=1}^{n_U}, \{x_j^P\}_{j=1}^{n_P})$, where the unlabeled samples $\{x_i^U\}_{i=1}^{n_U}$ are drawn i.i.d. from $p(x)$ over $\mathcal{X}$, and the positive samples $\{x_j^P\}_{j=1}^{n_P}$ are drawn i.i.d. from $p(x \mid y = +1)$ over $\mathcal{X}_P$.

We then verify the bounded difference condition for $\Phi$. Since $\ell : \mathbb{R} \to [0, B]$ is $L$-Lipschitz continuous, the composition $\ell(-f)$ inherits the Lipschitz property with constant $L$, and $\ell(-f(x)) \in [0, B]$ for all $x$. Consider replacing a single unlabeled sample $x_i^U$ with $\tilde{x}_i^U$, where both are drawn from $p(x)$ over $\mathcal{X}$, while keeping all other samples fixed. For any fixed $f \in \mathcal{H}$, the change in the unlabeled term of the empirical risk is bounded by

$$
\frac{1}{n_U} \left| \ell(-f(x_i^U)) - \ell(-f(\tilde{x}_i^U)) \right| \le \frac{B}{n_U}.
\tag{20}
$$

Similarly, consider replacing a single positive sample $x_j^P$ with $\tilde{x}_j^P$, where both are drawn from $p(x \mid y = +1)$ over $\mathcal{X}_P$. Since the uniform range bound of $g_f$ over $\mathcal{X}_P$ is $2B$, the change in the positive term of the empirical risk satisfies

$$
\frac{\pi}{n_P} \left| g_f(x_j^P) - g_f(\tilde{x}_j^P) \right| \le \frac{2\pi B}{n_P}.
\tag{21}
$$

The supremum over $\mathcal{H}$ inherits these bounded difference properties. To see this, for any $\epsilon > 0$, there exists $f \in \mathcal{H}$ such that $\Phi \le |\widehat{R}_{\mathrm{PU}}(f) - R_{\mathrm{PU}}(f)| + \epsilon$. Let $\Phi'$ denote the supremum after replacing one sample. It follows that

$$
\begin{aligned}
\Phi - \Phi' &\le |\widehat{R}_{\mathrm{PU}}(f) - R_{\mathrm{PU}}(f)| + \epsilon - |\widehat{R}_{\mathrm{PU}}'(f) - R_{\mathrm{PU}}(f)| \\
&\le |\widehat{R}_{\mathrm{PU}}(f) - \widehat{R}_{\mathrm{PU}}'(f)| + \epsilon.
\end{aligned}
\tag{22}
$$

Taking $\epsilon \to 0$ and applying the same argument with the roles interchanged, we obtain that $|\Phi - \Phi'|$ is bounded by the maximum change in empirical risk over all $f \in \mathcal{H}$.

By McDiarmid's inequality applied to $\Phi$ as a function of the $n_U + n_P$ samples, we have

$$
\Pr\{\Phi - \mathbb{E}[\Phi] \ge t\} \le \exp\left( -\frac{2t^2}{\sum_{i=1}^{n_U}(B/n_U)^2 + \sum_{j=1}^{n_P}(2\pi B/n_P)^2} \right).
\tag{23}
$$

The denominator is simplified as

$$
n_U \cdot \frac{B^2}{n_U^2} + n_P \cdot \frac{4\pi^2 B^2}{n_P^2} = \frac{B^2}{n_U} + \frac{4\pi^2 B^2}{n_P}.
\tag{24}
$$

Setting the right-hand side equal to $\delta/2$ yields the threshold

$$t = B\sqrt{\frac{\ln(2/\delta)}{2}\left(\frac{1}{n_{\mathrm{U}}} + \frac{4\pi^2}{n_{\mathrm{P}}}\right)}. \tag{25}$$

It remains to bound the expectation $\mathbb{E}[\Phi]$. By standard symmetrization arguments (Shalev-Shwartz & Ben-David, 2014), the expectation satisfies

$$\mathbb{E}[\Phi] \leq 2\mathbb{E}\left[\sup_{f\in\mathcal{H}}\left|\frac{1}{n_{\mathrm{U}}}\sum_{i=1}^{n_{\mathrm{U}}}\sigma_i\ell(-f(x_i^{\mathrm{U}}))\right|\right] + 2\pi\mathbb{E}\left[\sup_{f\in\mathcal{H}}\left|\frac{1}{n_{\mathrm{P}}}\sum_{j=1}^{n_{\mathrm{P}}}\sigma_j g_f(x_j^{\mathrm{P}})\right|\right], \tag{26}$$

where $\{\sigma_i\}$ and $\{\sigma_j\}$ are independent Rademacher random variables. By the contraction principle for Rademacher complexity (Ledoux & Talagrand, 2013), the first term is bounded as

$$\mathbb{E}\left[\sup_{f\in\mathcal{H}}\left|\frac{1}{n_{\mathrm{U}}}\sum_{i=1}^{n_{\mathrm{U}}}\sigma_i\ell(-f(x_i^{\mathrm{U}}))\right|\right] \leq L\cdot\mathfrak{R}_{n_{\mathrm{U}}}(\mathcal{H}). \tag{27}$$

Similarly, since $g_f$ is $2L$-Lipschitz as a function of $f$, we have

$$\mathbb{E}\left[\sup_{f\in\mathcal{H}}\left|\frac{1}{n_{\mathrm{P}}}\sum_{j=1}^{n_{\mathrm{P}}}\sigma_j g_f(x_j^{\mathrm{P}})\right|\right] \leq 2L\cdot\mathfrak{R}_{n_{\mathrm{P}}}(\mathcal{H}), \tag{28}$$

implying that $\mathbb{E}[\Phi] \leq 2L\cdot\mathfrak{R}_{n_{\mathrm{U}}}(\mathcal{H}) + 4\pi L\cdot\mathfrak{R}_{n_{\mathrm{P}}}(\mathcal{H})$.

Combining the concentration bound with the expectation bound, using $\sqrt{a+b} \leq \sqrt{a}+\sqrt{b}$ for nonnegative $a, b$, and assuming $\mathfrak{R}_n(\mathcal{H}) \leq \kappa/\sqrt{n}$, we obtain that with probability at least $1-\delta/2$,

$$\sup_{f\in\mathcal{H}}\left|\widehat{R}_{\mathrm{PU}}(f) - R_{\mathrm{PU}}(f)\right| \leq \frac{2L\kappa}{\sqrt{n_{\mathrm{U}}}} + \frac{4\pi L\kappa}{\sqrt{n_{\mathrm{P}}}} + B\sqrt{\frac{\ln(2/\delta)}{2n_{\mathrm{U}}}} + \pi B\sqrt{\frac{2\ln(2/\delta)}{n_{\mathrm{P}}}}. \tag{29}$$

We now analyze the performance of the empirical risk minimizer $\widehat{f} \in \arg\min_{f\in\mathcal{H}}\widehat{R}_{\mathrm{PU}}(f)$. Let $f^* \in \arg\min_{f\in\mathcal{H}} R_{\mathrm{PU}}(f)$ denote the expected risk minimizer within the hypothesis class. The standard excess risk decomposition yields

$$\begin{aligned}R_{\mathrm{PU}}(\widehat{f}) - R_{\mathrm{PU}}(f^*) &= \left[R_{\mathrm{PU}}(\widehat{f}) - \widehat{R}_{\mathrm{PU}}(\widehat{f})\right] + \left[\widehat{R}_{\mathrm{PU}}(\widehat{f}) - \widehat{R}_{\mathrm{PU}}(f^*)\right] + \left[\widehat{R}_{\mathrm{PU}}(f^*) - R_{\mathrm{PU}}(f^*)\right]\\ &\leq \left|R_{\mathrm{PU}}(\widehat{f}) - \widehat{R}_{\mathrm{PU}}(\widehat{f})\right| + 0 + \left|\widehat{R}_{\mathrm{PU}}(f^*) - R_{\mathrm{PU}}(f^*)\right|\\ &\leq 2\sup_{f\in\mathcal{H}}\left|\widehat{R}_{\mathrm{PU}}(f) - R_{\mathrm{PU}}(f)\right|,\end{aligned} \tag{30}$$

where the second inequality uses $\widehat{R}_{\mathrm{PU}}(\widehat{f}) \leq \widehat{R}_{\mathrm{PU}}(f^*)$ by definition of the empirical risk minimizer.

Substituting the uniform deviation bound, with probability at least $1-\delta$

$$R_{\mathrm{PU}}(\widehat{f}) - R_{\mathrm{PU}}(f^*) \leq \frac{4L\kappa}{\sqrt{n_{\mathrm{U}}}} + 2B\sqrt{\frac{2\ln(2/\delta)}{n_{\mathrm{U}}}} + \frac{8\pi L\kappa}{\sqrt{n_{\mathrm{P}}}} + 2\pi B\sqrt{\frac{2\ln(2/\delta)}{n_{\mathrm{P}}}}. \tag{31}$$

Define the unlabeled-dependent constant $\Xi_{\mathrm{U}} = \frac{4L\kappa}{\sqrt{n_{\mathrm{U}}}} + 2B\sqrt{\frac{2\ln(2/\delta)}{n_{\mathrm{U}}}}$. To achieve excess risk at most $\epsilon$, we require

$$\frac{8\pi L\kappa + 2\pi B\sqrt{2\ln(2/\delta)}}{\sqrt{n_{\mathrm{P}}}} \leq \epsilon - \Xi_{\mathrm{U}}. \tag{32}$$

Solving for $n_{\mathrm{P}}$, the sufficient sample threshold satisfies

$$P^*_{\epsilon,\delta;\mathrm{PU}} = \left(\frac{8\pi L\kappa + 2\pi B\sqrt{2\ln(2/\delta)}}{\epsilon - \Xi_{\mathrm{U}}}\right)^2. \tag{33}$$

This bound is valid provided that $\epsilon > \Xi_{\mathrm{U}}$.

# B. Proof of Theorem 3.4

The proof establishes that variance regularization restricts the hypothesis class to $\mathcal{H}_\tau \subset \mathcal{H}$, where the Rademacher complexity is reduced from $\kappa$ to $\kappa_\tau \leq \kappa$. This reduction improves the complexity term in the generalization bound, while the concentration term governed by the range bound $2B$ remains unchanged.

Recall from the main text the restricted hypothesis class induced by the variance constraint

$$\mathcal{H}_\tau = \{f \in \mathcal{H} : \mathrm{Var}_{\mathrm{P}}[g_f] \leq \tau\} \subset \mathcal{H}, \tag{34}$$

where $\mathrm{Var}_{\mathrm{P}}[g_f] = \mathbb{E}_{x \sim p(x|y=+1)}[(g_f(x) - \mathbb{E}_{\mathrm{P}}[g_f])^2]$ denotes the variance of $g_f$ under the positive class distribution, and $g_f(x) = \ell(f(x)) - \ell(-f(x))$ is defined as in the main text.

By Lemma 3.3, the Rademacher complexity of the variance-constrained class satisfies $\mathfrak{R}_{n_{\mathrm{P}}}(\mathcal{G}_\tau) \leq \kappa_\tau/\sqrt{n_{\mathrm{P}}}$ with $\kappa_\tau \leq \kappa$.

The bounded difference condition for McDiarmid's inequality depends on the range, not variance. For any $f \in \mathcal{H}_\tau$, we still have $g_f(x) \in [-B, B]$, so the uniform range bound remains $2B$. This is because variance measures the spread around the mean, but does not constrain the extreme values that determine the range. Therefore, the concentration term in McDiarmid's inequality remains unchanged at $\pi B \sqrt{2 \ln(2/\delta)/n_{\mathrm{P}}}$.

We repeat the McDiarmid analysis from Theorem 3.2 over the restricted class $\mathcal{H}_\tau$ by defining

$$\Phi_\tau = \sup_{f \in \mathcal{H}_\tau} \left| \widehat{R}_{\mathrm{PU}}(f) - R_{\mathrm{PU}}(f) \right|. \tag{35}$$

The bounded differences are: $B/n_{\mathrm{U}}$ for the unlabeled term and $2\pi B/n_{\mathrm{P}}$ for the positive term, the same as in Theorem 3.2. By McDiarmid's inequality, we have

$$\Pr\{\Phi_\tau - \mathbb{E}[\Phi_\tau] \geq t\} \leq \exp\left(-\frac{2t^2}{\frac{B^2}{n_{\mathrm{U}}} + \frac{4\pi^2 B^2}{n_{\mathrm{P}}}}\right). \tag{36}$$

The expectation bound uses the reduced Rademacher complexity of $\mathcal{H}_\tau$. By the same symmetrization and contraction arguments as in Theorem 3.2, we obtain

$$\mathbb{E}[\Phi_\tau] \leq 2L \cdot \mathfrak{R}_{n_{\mathrm{U}}}(\mathcal{H}) + 4\pi L \cdot \mathfrak{R}_{n_{\mathrm{P}}}(\mathcal{H}_\tau) \leq \frac{2L\kappa}{\sqrt{n_{\mathrm{U}}}} + \frac{4\pi L \kappa_\tau}{\sqrt{n_{\mathrm{P}}}}. \tag{37}$$

Let $\widehat{f}_\tau$ denote the solution to the variance-constrained problem Eq.(8) with variance budget $\tau$. By standard Lagrangian duality (Boyd & Vandenberghe, 2004), for any $\tau > 0$, there exists a corresponding $\lambda > 0$ such that solving the Lagrangian problem Eq.(9) yields a solution in $\mathcal{H}_\tau$.

Combining the bounds and following the excess risk decomposition as in Theorem 3.2, with probability at least $1 - \delta$

$$R_{\mathrm{PU}}(\widehat{f}_\tau) - R_{\mathrm{PU}}(f^*) \leq \Xi_{\mathrm{U}} + \frac{8\pi L \kappa_\tau}{\sqrt{n_{\mathrm{P}}}} + 2\pi B \sqrt{\frac{2\ln(2/\delta)}{n_{\mathrm{P}}}}. \tag{38}$$

Solving for the minimal $n_{\mathrm{P}}$ yields the learning threshold

$$P^*_{\epsilon,\delta;\mathrm{PUVR}}(\tau) = \left(\frac{8\pi L \kappa_\tau + 2\pi B \sqrt{2\ln(2/\delta)}}{\epsilon - \Xi_{\mathrm{U}}}\right)^2. \tag{39}$$

The improvement over standard PU learning comes from $\kappa_\tau < \kappa$ in the complexity term $8\pi L \kappa_\tau$. The concentration term $2\pi B \sqrt{2\ln(2/\delta)}$ remains unchanged because variance regularization does not reduce the range bound.

## C. Proof of Lemma 3.3

We establish that the variance constraint $\mathrm{Var}_P[g_f] \leq \tau$ induces a restricted function class $\mathcal{G}_\tau = \{g_f : f \in \mathcal{H}_\tau\}$ with reduced Rademacher complexity parameter $\kappa_\tau \leq \kappa$.

We proceed via Dudley's entropy integral (Dudley, 1967). For the original class $\mathcal{G} = \{g_f : f \in \mathcal{H}\}$, the Rademacher complexity satisfies

$$\mathfrak{R}_{n_\mathrm{P}}(\mathcal{G}) \leq \int_0^B \sqrt{\frac{\log \mathcal{N}(\mathcal{G}, \|\cdot\|_{L_2(P)}, \epsilon)}{n_\mathrm{P}}}\, d\epsilon, \tag{40}$$

where $\mathcal{N}(\mathcal{G}, \|\cdot\|_{L_2(P)}, \epsilon)$ denotes the $\epsilon$-covering number of $\mathcal{G}$ under the $L_2(P)$ metric, and the integration runs over $[0, B]$ since $g_f(x) \in [-B, B]$ for all $f \in \mathcal{H}$.

For the variance-constrained class $\mathcal{G}_\tau$, we show that the effective integration range shrinks. Since $\mathrm{Var}_P[g_f] \leq \tau$ for all $g_f \in \mathcal{G}_\tau$, any function in $\mathcal{G}_\tau$ satisfies

$$\mathbb{E}_P\left[(g_f(x) - \mathbb{E}_P[g_f])^2\right] \leq \tau, \tag{41}$$

which implies $\|g_f - \mathbb{E}_P[g_f]\|_{L_2(P)} \leq \sqrt{\tau}$. Therefore, the $L_2(P)$ diameter of $\mathcal{G}_\tau$ around its mean is at most $\mathcal{O}(\sqrt{\tau})$, shrinking the effective integration range from $[0, B]$ to $[0, \mathcal{O}(\sqrt{\tau})]$. Applying Dudley's entropy integral to $\mathcal{G}_\tau$ yields

$$\mathfrak{R}_{n_\mathrm{P}}(\mathcal{G}_\tau) \lesssim \int_0^{\mathcal{O}(\sqrt{\tau})} \sqrt{\frac{\log \mathcal{N}(\mathcal{G}_\tau, \|\cdot\|_{L_2(P)}, \epsilon)}{n_\mathrm{P}}}\, d\epsilon. \tag{42}$$

Since $\mathcal{G}_\tau \subseteq \mathcal{G}$, the covering number satisfies $\mathcal{N}(\mathcal{G}_\tau, \|\cdot\|_{L_2(P)}, \epsilon) \leq \mathcal{N}(\mathcal{G}, \|\cdot\|_{L_2(P)}, \epsilon)$ monotonically. Combining this with the shrunken integration limit gives

$$\mathfrak{R}_{n_\mathrm{P}}(\mathcal{G}_\tau) \leq \frac{\kappa_\tau}{\sqrt{n_\mathrm{P}}}, \tag{43}$$

where $\kappa_\tau \leq \kappa$ follows from both the reduced upper limit of integration and the monotone decrease of the covering number. For sufficiently small $\tau$, the reduction is strict, i.e., $\kappa_\tau < \kappa$, since the shrunken integration range $[0, \mathcal{O}(\sqrt{\tau})]$ yields a strictly smaller integral than $[0, B]$ whenever $\tau < B^2$.

## D. Proof of Theorem 3.5

We establish that geometric compactness in the representation space reduces the effective range, thereby improving the concentration term in the generalization bound. Combined with variance regularization that reduces the complexity term, we obtain improvements to both components.

Since $\ell$ is $L$-Lipschitz and the classifier is parameterized as $f(z) = w^\top z + b$ with $\|w\|_2 \leq W$ (ensured by weight decay regularization), we have

$$
\begin{aligned}
|g_{f,h}(x) - g_{f,h}(x')| &= |[\ell(f(h(x))) - \ell(-f(h(x)))] - [\ell(f(h(x'))) - \ell(-f(h(x')))]| \\
&\leq |\ell(f(h(x))) - \ell(f(h(x')))| + |\ell(-f(h(x))) - \ell(-f(h(x')))| \\
&\leq L|f(h(x)) - f(h(x'))| + L|f(h(x)) - f(h(x'))| \\
&= 2L|w^\top(h(x) - h(x'))| \\
&\leq 2L\|w\|_2\|h(x) - h(x')\| \\
&\leq 2LW\|h(x) - h(x')\|.
\end{aligned} \tag{44}
$$

Thus $g_{f,h}$ is $L_g$-Lipschitz in the feature space with $L_g = 2LW$.

Let $\mu_\mathrm{P} = \mathbb{E}_{x \sim p(x|y=+1)}[h(x)]$ denote the empirical positive class prototype computed from the observed positive samples. For any $x, x' \in \mathcal{X}_\mathrm{P}$, we have

$$|g_{f,h}(x) - g_{f,h}(x')| \leq L_g\|h(x) - h(x')\| \leq L_g\left(\|h(x) - \mu_\mathrm{P}\| + \|\mu_\mathrm{P} - h(x')\|\right). \tag{45}$$

Recall from the main text the geometrically constrained class

$$\mathcal{H}_\rho^{\text{geo}} = \left\{ (f,h) : \mathbb{E}_{x \sim p(x|y=+1)}[\|h(x) - \mu_P\|^2] \leq \rho^2 \right\}. \tag{46}$$

For any $(f,h) \in \mathcal{H}_\rho^{\text{geo}}$, if positive representations lie within distance $\rho$ of the centroid, the range over $\mathcal{X}_P$ satisfies $\text{range}_P(g_{f,h}) \leq 2L_g\rho$.

The combined objective Eq.(16) simultaneously enforces two constraints that restrict the hypothesis class. Variance regularization with parameter $\lambda$ encourages solutions in $\mathcal{H}_\tau$ with reduced Rademacher complexity $\kappa_\tau \leq \kappa$. Geometric compactness with parameter $\gamma$ encourages solutions in $\mathcal{H}_\rho^{\text{geo}}$ with reduced effective range $2L_g\rho$.

We apply the McDiarmid analysis over the doubly-restricted hypothesis class $\mathcal{H}_\tau \cap \mathcal{H}_\rho^{\text{geo}}$. Define

$$\Phi_{\tau,\rho} = \sup_{(f,h) \in \mathcal{H}_\tau \cap \mathcal{H}_\rho^{\text{geo}}} \left| \widehat{R}_{\text{PU}}(f \circ h) - R_{\text{PU}}(f \circ h) \right|. \tag{47}$$

The bounded differences for the unlabeled term remain $B/n_U$. For the positive term, since the effective range of $g_{f,h}$ over $\mathcal{X}_P$ is at most $2L_g\rho$, the bounded difference becomes

$$\frac{\pi}{n_P} \left| g_{f,h}(x_j^P) - g_{f,h}(\tilde{x}_j^P) \right| \leq \frac{2\pi L_g \rho}{n_P}. \tag{48}$$

By McDiarmid's inequality, we have

$$\Pr\left\{ \Phi_{\tau,\rho} - \mathbb{E}[\Phi_{\tau,\rho}] \geq t \right\} \leq \exp\left( -\frac{2t^2}{\frac{B^2}{n_U} + \frac{4\pi^2 L_g^2 \rho^2}{n_P}} \right). \tag{49}$$

Setting the right-hand side equal to $\delta/2$ gives the deviation threshold

$$t = \sqrt{\frac{\ln(2/\delta)}{2} \left( \frac{B^2}{n_U} + \frac{4\pi^2 L_g^2 \rho^2}{n_P} \right)}. \tag{50}$$

The expectation bound uses the reduced Rademacher complexity from variance regularization

$$\mathbb{E}[\Phi_{\tau,\rho}] \leq \frac{2L\kappa}{\sqrt{n_U}} + \frac{4\pi L\kappa_\tau}{\sqrt{n_P}}. \tag{51}$$

Combining all bounds and following the excess risk decomposition, with probability at least $1 - \delta$

$$R_{\text{PU}}(\widehat{f} \circ \widehat{h}) - R_{\text{PU}}(f^*) \leq \Xi_U + \frac{8\pi L\kappa_\tau}{\sqrt{n_P}} + 2\pi L_g \rho \sqrt{\frac{2\ln(2/\delta)}{n_P}}. \tag{52}$$

Solving for the minimal $n_P$ yields the learning threshold for ScalePU as

$$P_{\epsilon,\delta;\text{ScalePU}}^*(\tau, \rho) = \left( \frac{8\pi L\kappa_\tau + 2\pi L_g \rho \sqrt{2\ln(2/\delta)}}{\epsilon - \Xi_U} \right)^2. \tag{53}$$

Compared to Theorem 3.2, the complexity term is reduced from $8\pi L\kappa$ to $8\pi L\kappa_\tau$ through variance regularization, and the concentration term is reduced from $2\pi B\sqrt{2\ln(2/\delta)}$ to $2\pi L_g \rho \sqrt{2\ln(2/\delta)}$ through geometric regularization. When $\kappa_\tau < \kappa$ and $L_g\rho < B$, we have $P_{\epsilon,\delta;\text{ScalePU}}^* < P_{\epsilon,\delta;\text{PU}}^*$, demonstrating that the two regularization mechanisms provide complementary improvements to different components of the generalization bound.

# E. Detailed Datasets

In this appendix, we provide detailed descriptions of all PU datasets in Table 1. We use eight diverse datasets covering both standard benchmarks and domain-specific applications. Specifically, we employ four widely-used image datasets: **CIFAR-10** (Krizhevsky & Hinton, 2009), **Fashion-MNIST** (F-MNIST) (Xiao et al., 2017), **STL-10** (Coates et al., 2011), and **ImageNette** (Deng et al., 2009), along with the **Alzheimer** MRI dataset. To create binary PU learning scenarios from multi-class datasets, we partition the original categories into positive and negative groups, following established conventions in the literature (Kiryo et al., 2017; Wang et al., 2023).

We adopt dataset-specific architectures following established practices: a 7-layer CNN (CNN-CIFAR) for CIFAR-10, the STL-adapted 7-layer CNN variant (CNN-STL) for STL-10, LeNet-5 (LeCun et al., 1998) for F-MNIST, and ResNet-50 (He et al., 2016) for high-resolution datasets (*i.e.,* ImageNette, Alzheimer). All models are trained using Adam (Kingma & Ba, 2014). Batch sizes are adapted based on $n_{\mathrm{P}}$. Training proceeds for 100 epochs with the sigmoid surrogate loss.

# F. Detailed Baseline Methods

In this appendix, we provide detailed descriptions of all baseline methods used in our experiments. These methods encompass two major paradigms in PU learning: disambiguation-free empirical risk estimators and pseudo-labeling methods.

## F.1. Disambiguation-Free Empirical Risk Estimators

These methods directly estimate the classification risk from positive and unlabeled data without explicitly refining labels for unlabeled samples.

**uPU** (**U**nbiased **PU** Learning) (du Plessis et al., 2015). This foundational method derives an unbiased risk estimator by decomposing the expected risk over labeled positive and unlabeled distributions. The risk is expressed as:

$$\widehat{R}_{\mathrm{uPU}}(f) = \pi \widehat{R}_P^+(f) + \widehat{R}_U^-(f) - \pi \widehat{R}_P^-(f), \tag{54}$$

where $\widehat{R}_P^+$ and $\widehat{R}_P^-$ denote empirical risks on positive data for positive and negative predictions respectively, and $\widehat{R}_U^-$ is the empirical risk on unlabeled data. While theoretically unbiased, this estimator can produce negative values during training, leading to severe overfitting when the model is flexible.

**nnPU** (**N**on-**N**egative **PU** Learning) (Kiryo et al., 2017). To address the overfitting issue of uPU, nnPU introduces a non-negativity constraint on the negative risk component:

$$\widehat{R}_{\mathrm{nnPU}}(f) = \pi \widehat{R}_P^+(f) + \max\left\{0, \widehat{R}_U^-(f) - \pi \widehat{R}_P^-(f)\right\}. \tag{55}$$

When the estimated negative risk becomes negative (indicating overfitting), the gradient is set to zero, preventing the model from exploiting spurious patterns.

**abs-PU** (**Abs**olute Value **PU** Learning) (Hammoudeh & Lowd, 2020). This method proposes an alternative correction mechanism using absolute values:

$$\widehat{R}_{\mathrm{abs}}(f) = \pi \widehat{R}_P^+(f) + \left| \widehat{R}_U^-(f) - \pi \widehat{R}_P^-(f) \right|. \tag{56}$$

The absolute value ensures non-negativity while maintaining gradient flow even when the original estimate is negative. But it may introduce bias when the negative risk estimate fluctuates around zero.

**DC-PU** (**D**ual-**C**onstrained **PU** learning) (Li et al., 2025). DC-PU extends PU learning to consider fairness constraints with respect to balanced error rates between the positive and negative classes by introducing weak and strong equality constraints on them:

$$\min_g \ \pi \widehat{R}_p^+(g) + \max\left\{\omega, \widehat{R}_u^-(g) - \pi \widehat{R}_p^-(g)\right\} \qquad \textbf{s.t.} \ \ \widehat{R}_p^+(g) = \frac{\widehat{R}_u^-(g) - \pi \widehat{R}_p^-(g)}{1 - \pi} \tag{57}$$

where $\omega$ is dynamically updated during classifier training.

### F.2. Pseudo-Labeling Methods

These methods assign pseudo-labels to unlabeled samples and iteratively refine predictions through self-training mechanisms.

**VPU** (**V**ariational **PU** Learning) (Chen et al., 2020a). VPU introduces a variational framework for PU learning that models the uncertainty in pseudo-labels. It employs a variational autoencoder structure to learn latent representations and uses the reconstruction error as an auxiliary signal for identifying likely negative samples, where the variational terms help regularize the latent space and improve pseudo-label quality.

**P³Mix** (**P**U Learning with **P**artially **P**ositive **Mix**up) (Li et al., 2022). This method extends the Mixup (Zhang et al., 2018) data augmentation technique to PU learning by carefully selecting mixing partners. P³Mix identifies likely positive samples from the unlabeled set based on prediction confidence and uses them as preferential mixing partners.

**HolisticPU** (**PU** Learning through **Holistic** Predictive Trends) (Wang et al., 2023). HolisticPU takes a two-phase approach that exploits holistic predictive trends during training. In the warming phase, it trains a standard classifier to observe prediction trajectories for each unlabeled sample. Samples whose predictions consistently trend toward positive are identified as likely positives. In the fine-tuning phase, these identified samples are incorporated with soft labels based on their trend scores.

**LaGAM** (**La**tent **G**roup-**A**ware **M**eta-disambiguation) (Long et al., 2024). LaGAM addresses PU learning through meta-learning with latent group discovery. It assumes that unlabeled samples can be partitioned into latent groups with varying positive proportions. The method learns to identify these groups and assigns group-specific weights through a meta-learning objective. The inner optimization learns group assignments while the outer optimization trains the classifier with group-aware reweighting.

**PUL-CPBF** (**PU** Learning with **C**ontrolled **P**robability **B**oundary **F**ence) (Li et al., 2024). This recent method focuses on controlling the decision boundary through probability calibration.

## G. Implementation Details

**Hyperparameter Settings.** For specific hyperparameters, these values are selected based on the parameter sensitivity analysis in Section 4.4, which shows that performance is robust within the ranges $\lambda \in [10^{-1}, 10^0]$, $\gamma \in [10^{-2}, 10^0]$, and $\beta \in [10^{-1}, 10^1]$.

**Rademacher complexity parameter $\kappa$.** The parameter $\kappa$ characterizes the capacity of the hypothesis class through $\mathfrak{R}_n(\mathcal{H}) \leq \kappa/\sqrt{n}$. For linear models in $\mathbb{R}^m$, the Rademacher complexity satisfies (Bartlett & Mendelson, 2002)

$$\mathfrak{R}_n(\mathcal{H}) \leq \frac{\|W\|_F \cdot \|\mathbf{F}\|_2}{\sqrt{n}} \tag{58}$$

where $\|W\|_F$ is the Frobenius norm of the weight matrix (Krogh & Hertz, 1991) and $\|\mathbf{F}\|_2$ is the spectral norm of the feature covariance. For neural networks with $L$ layers, we define the parameter as

$$\kappa := \frac{\prod_{i=1}^{L} \|W_i\|_2 \cdot \sqrt{\sum_{i=1}^{L} \|W_i\|_F^2}}{\sqrt{m}} \cdot \|\mathbf{F}\|_2 \tag{59}$$

where $m$ is the input dimension. In practice, $\|\mathbf{F}\|_2$ is estimated from unlabeled data.

**Relationship between regularization and hypothesis class shrinkage.** The variance regularization term $\Omega_{\text{var}}(f)$ enforces a constraint on the expected functional variance over the positive distribution, thereby inducing a restricted hypothesis class $\mathcal{H}_\tau \subset \mathcal{H}$. This mechanism effectively prunes high-complexity functions that exhibit excessive fluctuations on $\mathcal{X}_P$, which leads to a reduced Rademacher complexity parameter $\kappa_\tau \leq \kappa$ and consequently lowers the complexity term in the generalization bound. Complementarily, the geometric regularizer $\Omega_{\text{compact}}(h)$ restricts the hypothesis space to $\mathcal{H}_\rho^{\text{geo}}$ by enforcing representation compactness in the feature space. By leveraging the Lipschitz continuity of $g_f$, this geometric constraint allows us to replace the loose uniform range bound $B$ with a significantly tighter effective range parameter $L_g\rho$, directly shrinking the concentration term in the sufficient sample threshold.

## H. Algorithm

We summarize the complete training procedure for ScalePU in Algorithm 1. To maintain the formal consistency of the sample sets without modifying the main text, we let $\mathcal{D}_P = \{x_i^P\}_{i=1}^{n_P}$ and $\mathcal{D}_U = \{x_j^U\}_{j=1}^{n_U}$ denote the sets of positive and unlabeled examples, respectively.

---

**Algorithm 1** ScalePU: PU Learning with Variance and Geometric Regularization

---

**Input:** Positive examples $\mathcal{D}_P$, unlabeled examples $\mathcal{D}_U$, class prior $\pi$, hyperparameters $\lambda, \gamma, \beta$, margin $m$, kernel bandwidth $\sigma$.
**Output:** Trained classifier $f \circ h$.
1: Initialize feature extractor $h_\phi$ and classifier $f_\theta$.
2: **while** not converged **do**
3:     Sample mini-batches $\mathcal{B}_P \subset \mathcal{D}_P$ and $\mathcal{B}_U \subset \mathcal{D}_U$.
4:     *// Compute positive prototype*
5:     $c_P \leftarrow \frac{1}{|\mathcal{B}_P|} \sum_{x_i^P \in \mathcal{B}_P} h(x_i^P)$.
6:     *// Compute PU risk Eq.(1)*
7:     $\widehat{R}_{PU} \leftarrow \frac{1}{|\mathcal{B}_U|} \sum_{x_j^U \in \mathcal{B}_U} \ell(-f(h(x_j^U))) + \frac{\pi}{|\mathcal{B}_P|} \sum_{x_i^P \in \mathcal{B}_P} g_i$.
8:     where $g_i = \ell(f(h(x_i^P))) - \ell(-f(h(x_i^P)))$.
9:     *// Compute variance regularization Eq.(10)*
10:     $\bar{g} \leftarrow \frac{1}{|\mathcal{B}_P|} \sum_i g_i$.
11:     $\Omega_{var} \leftarrow \pi \cdot \frac{1}{|\mathcal{B}_P|} \sum_i (g_i - \bar{g})^2$.
12:     *// Compute geometric regularization (Eqs. 14-15)*
13:     $\Omega_{compact} \leftarrow \frac{1}{|\mathcal{B}_P|} \sum_{x_i^P \in \mathcal{B}_P} \|h(x_i^P) - c_P\|^2$.
14:     $w_j \leftarrow 1 - \exp(-\|h(x_j^U) - c_P\|^2/\sigma^2)$ for $x_j^U \in \mathcal{B}_U$.
15:     $\Omega_{sep} \leftarrow \frac{1}{|\mathcal{B}_U|} \sum_{x_j^U \in \mathcal{B}_U} w_j \cdot \max(0, m - \|h(x_j^U) - c_P\|^2)$.
16:     *// Compute total loss Eq.(16)*
17:     $\mathcal{L} \leftarrow \widehat{R}_{PU} + \lambda \cdot \Omega_{var} + \gamma \cdot (\Omega_{compact} + \beta \cdot \Omega_{sep})$.
18:     Update $(\theta, \phi) \leftarrow (\theta, \phi) - \eta \nabla_{(\theta, \phi)} \mathcal{L}$.
19: **end while**

---

## I. Relationship to Semi-Supervised Learning

Although PU learning and semi-supervised learning (SSL) (Berthelot et al., 2019; Sohn et al., 2020) both exploit unlabeled data, they differ fundamentally in structure. Under the SCAR assumption, the unlabeled set in PU learning is a mixture of positive and negative examples drawn from the marginal distribution $p(x)$, functioning as a noisy negative set rather than truly unlabeled data. This distinction gives rise to the corrective term $g_f(x) = \ell(f(x)) - \ell(-f(x))$ in the PU risk estimator, which has no counterpart in SSL since unlabeled data in SSL carry no class-conditional contamination structure. As a direct consequence of this asymmetry, the unlabeled term in the PU risk estimator remains well-estimated for any large $n_U$, while the positive term becomes the sole source of estimation instability as $n_P \to 0$. This is precisely why $\kappa$ and $B$, as structural parameters of the positive term, are the correct targets for regularization in our setting, rather than properties of a symmetric joint risk as in SSL.

