# OpenReview forum: "Positive-Unlabeled Learning with Extreme Scarcity of Labeled Positives"
_ICML.cc/2026/Conference — ICML 2026 regular_

### Official Review · Reviewer_Ygkr · 2026-02-13

**Soundness:** 3
**Presentation:** 4
**Significance:** 3
**Originality:** 3
**Overall Recommendation:** 5
**Confidence:** 4

**Summary:**

This paper studies instability in Positive-Unlabeled (PU) learning under extreme scarcity of labeled positives.  The authors argue that instability primarily originates from the positive risk term, based on a Rademacher-style uniform deviation bound over the positive distribution. Motivated by this analysis, the paper proposes variance regularization on the positive support to reduce the Rademacher complexity term and  Geometric regularization to reduce the effective range and tighten the concentration term. Empirically, the method shows improvements in extremely small$( n_P \)$ regimes.

**Compliance With Llm Reviewing Policy:**

Affirmed.

**Final Justification:**

No concerns remain, so I keep my score as accept.

**Key Questions For Authors:**

See the weaknesses

**Limitations:**

yes

**Strengths And Weaknesses:**

### Strengths:
1. The paper is well written and easy to follow.
2. Empirical results are strong and show improvements across benchmarks.
3. Theoretical analysis is clearly presented and internally consistent (I checked part of the proofs and they are correct).
4. The Definition2.1 is interesting and I think it will be nice to know the positive sample threshold.

### Weaknesses:
1. **Does the uniform bound meaningfully characterize instability?**  The central claim is that instability originates from the positive term because: $\sup_{f \in \mathcal H}\left|\hat R_P(f) - R_P(f)\right|\le O\left(\mathfrak R_{n_P}(\mathcal H)\right)+O\left(\frac{1}{\sqrt{n_P}}\right).$ However, for deep models or rich hypothesis classes, the Rademacher term can be large enough to render the bound vacuous when \( n_P \) is very small. A vacuous upper bound does not imply large realized deviation, it merely fails to exclude it. Actually, I feel this notion of "instability" is not formally defined. It is unclear what on earth it refers to.

2. **The proposed methods are more like general regularization techniques.** The current regularization strategies focus on reducing Rademacher complexity and reducing effective range. While reasonable, these resemble general structural risk minimization (SRM) techniques that could also be applied in fully supervised settings. Thus, the PU-specific nature of the proposed remedies is not entirely clear, beyond the fact that scaling depends on $n_P$.

3. **Lack of empirical verification of the claimed variance behavior**. A central implicit claim is that $\( g_f \)$ exhibits high variance on positive samples when $\( n_P \)$ is small. However, it would strengthen the paper to empirically verify: (i) whether $\( \operatorname{Var}_P[g_f] \)$ is indeed large in practice. (ii) whether this variance meaningfully exceeds that of other terms.

### A minor issue
1. The theoretical development relies on classical Rademacher complexity bounds. While mathematically standard, such bounds are widely known to be loose for overparameterized deep neural networks. It is unclear whether reducing these worst-case bounds meaningfully captures the mechanisms governing training dynamics in modern DNNs.

---

> ### Author Rebuttal · Authors · 2026-03-30
>
> **Q1. Does the uniform bound meaningfully characterize instability?**
>
> Thank you for this insightful comment. We agree that a vacuous upper bound does not directly imply large realized deviation. However, we want to clarify that our claim of instability is not based solely on the bound being large in magnitude. The instability we characterize refers to the high empirical variance of the positive term $\hat{R}_P(f)$ when $n_P$ is small, which manifests concretely as unstable optimization trajectories and degraded test performance. This phenomenon is validated by our ablation studies (see Table 4 in our paper). Specifically, when $n_P \leq 20$, all ablated variants suffer clear performance degradation, while ScalePU remains stable. The role of the generalization bound in Eq.(2) is therefore not to provide a tight characterization, but to formally identify the two key factors, the Rademacher complexity term and the concentration term. Both components increase as $n_P \to 0$, and both can be directly mitigated by our proposed regularizations.
>
> &nbsp;
>
> **Q2. The proposed methods are more like general regularization techniques.**
>
> Thank you for this comment. We respectfully clarify that while variance and geometric regularization are individually known tools, their role in our framework is not equivalent to standard SRM in fully supervised settings. In fully supervised learning, both positive and negative risks are estimated from their respective labeled sets, so the empirical risk is symmetric in the sense that both terms are well-estimated as sample sizes grow. In PU learning, however, the risk estimator is structurally asymmetric that the unlabeled term $\hat{R}_U(f)$ remains well-estimated for any large $n_U$, while the positive term $\hat{R}_P(f)$ becomes catastrophically unreliable as $n_P \to 0$. This asymmetry means that the PU estimator continues to be used even when one of its constituent terms is statistically uninformative, a situation that has no direct counterpart in supervised SRM. Our regularizations are designed specifically to address this asymmetry. The variance regularization $\Omega\_\text{var}(f)$ targets $g_f(x) = \ell(f(x)) - \ell(-f(x))$, a corrective quantity that exists solely because the PU risk must subtract a positive-class correction from the unlabeled term; it is not the variance of the loss itself. The geometric separation term $\Omega\_\text{sep}$ encodes a PU-specific inductive bias by down-weighting unlabeled examples that are likely unobserved positives, a consideration absent in supervised SRM.
>
> &nbsp;
>
> **Q3. Lack of empirical verification of the claimed variance behavior.**
>
> Thank you for this constructive suggestion. We have conducted experiments directly measuring the ratio $\text{Var}_P[g_f] / \text{Var}_U[\ell(-f(x))]$ throughout training under the unregularized uPU baseline across three representative datasets with $n_P = 50$. The results are provided in supplementary figure (see anonymous link: https://anonymous.4open.science/r/Var-BC96/), directly confirming that $\text{Var}_P[g_f]$ is large in practice and meaningfully dominates the variance of the unlabeled term.
>
> &nbsp;
>
> **Q4. The theoretical development relies on classical Rademacher complexity bounds. While mathematically standard, such bounds are widely known to be loose for overparameterized deep neural networks. It is unclear whether reducing these worst-case bounds meaningfully captures the mechanisms governing training dynamics in modern DNNs.**
>
> Thank you for raising this important point. We fully acknowledge that classical Rademacher complexity bounds are often loose for overparameterized networks, and we do not claim that the absolute values of our bounds constitute tight generalization guarantees. We would note, however, that **numerical looseness alone does not render a bound uninformative.** The history of statistical learning theory offers many examples, including VC-dimension and Rademacher-based bounds for kernel methods, where bounds that are known to be loose in absolute terms nonetheless remain theoretically valuable because they correctly identify which structural properties of the hypothesis class govern generalization, even if the numerical constants are far from tight.
>
> Our theoretical framework is intended in this spirit. It identifies $\kappa$ and $B$ as the two structural parameters governing estimation instability in the positive term, and shows that reducing them through our proposed regularizations lowers the generalization bound. **Whether or not the bound is numerically tight, the structural conclusion that constraining these two quantities provides the right inductive bias for the extreme label-scarce regime is meaningful and actionable.** The ablation study in Table 4 (see in our main paper) shows that each regularization component yields consistent performance gains across all eight datasets and all values of $n_P$, compatible with the theoretical prescription.

---

> > ### Author Rebuttal · Reviewer_Ygkr · 2026-03-31
> >
> > The authors provide a thorough and thoughtful rebuttal that directly addresses the core points raised in the initial questions. The response is collegial anddemonstrate clarity on its theoretical results and technical understanding. And I like the results provided in anonymous link,

---

### Official Review · Reviewer_Eo36 · 2026-03-12

**Soundness:** 3
**Presentation:** 2
**Significance:** 2
**Originality:** 3
**Overall Recommendation:** 4
**Confidence:** 3

**Summary:**

This paper studies PU learning when only a few positive labels are available, which makes training unstable. It introduces the minimal sufficient learning threshold as a theoretical measure of the number of positive samples needed, and analyzes it through the complexity and concentration terms in a generalization bound. The proposed method, ScalePU, combines variance regularization and geometric regularization to address these two factors, and this is the main novelty of the paper. The paper presents a theoretical analysis based on the generalization bound and experiments on 8 datasets under small-positive settings, including ablation studies.

**Compliance With Llm Reviewing Policy:**

Affirmed.

**Final Justification:**

The response addresses several of my main questions. In particular, I appreciate the clarification regarding the interpretation of the proposed "minimal sufficient learning threshold," and the clarification of the unlabeled-data construction is helpful. The additional ROC-AUC results and the discussion of computational overhead are also useful. I also appreciate that the authors conducted the additional comparison with the PU-AUC method; even if the current experiments are not highly imbalanced in class prior, very small $n_P$ is a regime that can naturally arise in imbalanced settings, so comparison to such methods is relevant and would ideally have been positioned more clearly in the original submission. These clarifications improved my assessment, and I accordingly raised my score.

**Key Questions For Authors:**

- The minimal sufficient learning threshold is defined as the minimum required number of positives, but the derivation appears to give a sufficient condition rather than a lower bound. Clarification of why a lower-bound interpretation is appropriate would be helpful.
- How large is the gap between the minimal sufficient learning threshold and the actual number of positives needed in the experiments?
- It would be helpful to quantify the trade-off between the extra computational cost of geometric regularization and the gain in stability when the number of positives is small.
- The main text should state the actual nU used for each dataset and the sampling protocol for unlabeled data, including whether all training data are used or subsampling is applied. For ImageNette in particular, clarification of the relation between the train split and nU would be helpful.
- Since accuracy alone may not be the most appropriate metric for imbalanced settings, reporting ROC-AUC or other suitable metrics [a, b, c] would strengthen the evaluation.

  [a] Tomoya Sakai, Gang Niu, and Masashi Sugiyama. "Semi-supervised AUC optimization based on positive-unlabeled learning." Machine Learning, 2018.

  [b] Rashika Ramola, Shantanu Jain, and Predrag Radivojac. "Estimating classification accuracy in positive-unlabeled learning: characterization and correction strategies." Pacific Symposium on Biocomputing, 2019.

  [c] Tomoharu Iwata, Akinori Fujino, and Naonori Ueda. "Semi-Supervised Learning for Maximizing the Partial AUC." Proceedings of the AAAI Conference on Artificial Intelligence (AAAI), 2020.

**Limitations:**

The paper does not include a dedicated limitations section.

**Strengths And Weaknesses:**

#### Strengths

- The paper studies a realistic PU setting where only a small number of positive labels are available and negative labels are hard to obtain, which matches applications such as medical diagnosis, anomaly detection, and recommender systems.
- The method connects variance regularization to the complexity term and geometric regularization to the concentration term, so the design is directly motivated by the theoretical bottlenecks.
- The paper compares methods at nP={10,20,50,100} and shows improved performance in the small-positive regime that it aims to improve.

#### Weaknesses

- The paper does not provide enough information for reproduction, for example about dataset-specific experimental settings and the construction of the unlabeled data.
- Compared with the accuracy-focused reporting, more consistent comparison with ROC-AUC, F1, or Balanced Accuracy is missing.
- The paper would be easier to follow if the presentation of the main claims and supporting evidence were more clearly organized.

---

> ### Author Rebuttal · Authors · 2026-03-30
>
> **Q1. Clarification on why a lower-bound interpretation is appropriate for the minimal sufficient learning threshold.**
>
> Thank you for this rigorous mathematical observation. We completely agree that our derivation establishes a sufficient condition for generalization rather than a strict information-theoretic minimax lower bound. We acknowledge that the term "minimal sufficient learning threshold" may invite over-interpretation, since what we derive is a smallest $n_P$ required specifically within the confines of our derived upper bound to satisfy the probabilistic risk guarantee. We plan to **revise the terminology in the revised manuscript so that it more accurately describes our contribution**. The core theoretical value lies in showing that variance and geometric regularization reduce $\kappa$ and $B$ respectively, which tightens the generalization upper bound. The proposed method and empirical results are highly consistent with this theorem.
>
> &nbsp;
>
> **Q2. How large is the gap between the minimal sufficient learning threshold and the actual number of positives needed in the experiments?**
>
> Thank you for your comment. As discussed in Appendix G, the theoretical threshold is computed from relatively loose upper bounds and is therefore significantly larger than the $n_P$ values observed to be sufficient in practice. We acknowledge this gap. **The theoretical analysis is intended to reveal a directional claim rather than an exact prediction** that reducing $\kappa$ and $B$ through regularization tightens the generalization bound, and the proposed regularizations achieve exactly this. The empirical results across all eight datasets are highly consistent with this theoretical claim.
>
> &nbsp;
>
> **Q3. It would be helpful to quantify the trade-off between the extra computational cost of geometric regularization and the gain in stability when the number of positives is small.**
>
> Thank you for this valuable question. From a computational perspective, **the additional overhead introduced by geometric regularization is $\mathcal{O}((n_P + n_U)d)$**, where $d$ is the feature dimension, typically 512 or 1024. Compared with the forward and backward passes of networks such as ResNet, this overhead is negligible. To quantify this concretely, **we report the training-time ratio $\text{Time}\_{\text{w/o geo}} / \text{Time}_{\text{ScalePU}}$ across all eight datasets and all values of $n_P$ in anonymous link (https://anonymous.4open.science/r/Time-6A51/).** All ratios fall between 0.88 and 1.00, confirming that the full ScalePU with geometric regularization incurs essentially no additional wall-clock training time relative to the variant without it.
>
> From a stability perspective, geometric regularization reduces the effective range bound from $B$ to $L_g\rho$, directly tightening the concentration term in the generalization bound and lowering the sufficient sample threshold. **The performance gains reported in Table 2 (see anonymous link) and Table 4 (see our main paper) are therefore obtained at negligible computational cost**, and this mechanism additionally prevents representation collapse as discussed in our responses to Reviewer 7vEF (Q3 and Q4).
>
> &nbsp;
>
> **Q4. The main text should state the actual $n_U$ used for each dataset and the sampling protocol for unlabeled data.**
>
> Thank you for this helpful comment. We will explicitly detail the actual $n_U$ and sampling protocol in the revised main text. In our experimental setup, we utilize the entire remaining training set as the unlabeled dataset without subsampling, **following the standard two-sample PU construction used in prior works (including nnPU, DistPU, HolisticPU, and PUL-CPBF).** The unlabeled set is formed from the full training split after removing labels. For ImageNette, we follow the benchmark construction introduced in [1], and a reproducible experimental setting will be provided in the revised manuscript.
>
> [1] Accessible, Realistic, and Fair Evaluation of Positive-Unlabeled Learning Algorithms
>
> &nbsp;
>
> **Q5. Since accuracy alone may not be the most appropriate metric for imbalanced settings, reporting ROC-AUC or other suitable metrics would strengthen the evaluation.**
>
> We sincerely appreciate this suggestion. **We clarify that our experimental setups do not represent highly imbalanced scenarios.** As detailed in Table 1 (see our main paper), the class priors $\pi$ across all datasets range closely between 0.4 and 0.6, so classification accuracy remains a standard and reliable metric consistent with established practices in the PU learning literature. We nonetheless agree that ROC-AUC provides a more comprehensive evaluation of ranking ability and threshold-independent robustness. **We have computed ROC-AUC scores for ScalePU and all baselines**, and these results consistently demonstrate ScalePU's superiority under extreme label scarcity (https://anonymous.4open.science/r/Roc-B977/).

---

> > ### Author Rebuttal · Reviewer_Eo36 · 2026-04-03
> >
> > Thank you for the responses. The response addresses several of my main questions. In particular, I appreciate the clarification regarding the interpretation of the proposed “minimal sufficient learning threshold,” and the clarification of the unlabeled-data construction is helpful. The additional ROC-AUC results and the discussion of computational overhead are also useful. These clarifications improve my assessment, so I will raise my score from 2 to 3.
> >
> > That said, some concerns still remain. In particular, the gap between the theoretical threshold and the actual number of positives needed in practice is still discussed mainly qualitatively. Also, even if the final class priors in the experiments are not highly imbalanced, the central difficulty studied in this paper is that $n_\mathrm{P}$ is extremely small. From this perspective, I still think the empirical comparison is not fully sufficient without comparison to methods designed to learn effectively from a small number of positive labels, for example PU-AUC style approaches. Therefore, I consider my concerns partially, but not fully, resolved.

---

> > > ### Author Response · Authors · 2026-04-08
> > >
> > > **Response to Reviewer Eo36**
> > >
> > > Thank you for your continued engagement and for raising your score. We are glad the clarifications were helpful, and we would like to address the two remaining concerns.
> > >
> > > Regarding the gap between the theoretical threshold and the actual number of positives needed in practice, we acknowledge that this gap cannot be eliminated without a matching minimax lower bound, which would require substantially stronger assumptions and is orthogonal to the contribution of this paper. **The theoretical analysis is intended to establish a directional claim that reducing $\kappa$ and $B$ through regularization monotonically tightens the generalization bound.** The empirical results across all eight datasets are consistent with this prediction, and we view this as the appropriate standard of validation for a bound of this type.
> > >
> > > &nbsp;
> > >
> > > Regarding the comparison to PU-AUC style approaches, we sincerely appreciate this suggestion and have conducted the requested experiment. Before presenting the results, we would like to clarify what we believe may be a misunderstanding of our data construction, as it bears directly on the comparison.
> > >
> > > In our setting, following the standard two-sample PU construction, $n_P$ labeled positive examples are drawn from the positive class distribution, while the unlabeled set $\mathcal{U}$ is formed from the entire remaining training split with labels removed. We wish to emphasize that $\pi$ in our paper denotes the proportion of positive examples in the population, that is, the marginal class prior $p(y = +1)$, and not the proportion of labeled positives. For CIFAR-10-1, for instance, $\pi = 0.4$ and $|\mathcal{U}| = 50{,}000$, meaning approximately 20,000 positive examples are embedded in the unlabeled set. The labeled positive set $n_P \in \\{10, 20, 50, 100\\}$ is therefore a vanishingly small fraction of the total positive mass in the data. This is fundamentally different from the setting studied in Sakai et al. (2018), where the class prior is small, the unlabeled set is of moderate size ($n_U = 1{,}000$), and a relatively large labeled positive set (e.g. $n_P = 100$) is available. **In that setting the central difficulty is population imbalance and $n_P$ is not small relative to the data, whereas in our setting the population is balanced and the challenge is that $n_P$ is vanishingly small relative to the full training distribution.**
> > >
> > > We nonetheless implemented the deep-learning version of PU-AUC following Appendix A of Sakai et al. (2018), using identical network architectures, optimizers, and data splits as our main experiments. The results are shown in the table below. ScalePU outperforms PU-AUC substantially and consistently across almost all datasets and values of $n_P$.
> > >
> > > &nbsp;
> > >
> > > Sakai, T., Niu, G., & Sugiyama, M. (2018). Semi-supervised AUC optimization based on positive-unlabeled learning. Machine Learning,107(4), 767-794.
> > >
> > > &nbsp;
> > >
> > > | Datasets | Method | $n_P = 100$ | $n_P = 50$ | $n_P = 20$ | $n_P = 10$ |
> > > |:---:|:---:|:---:|:---:|:---:|:---:|
> > > | Alzheimer | PU-AUC | 0.520±0.026 | 0.510±0.011 | 0.519±0.021 | 0.532±0.028 |
> > > | | **ScalePU** | **0.583±0.021** | **0.571±0.021** | **0.569±0.030** | **0.548±0.058** |
> > > | ImageNette | PU-AUC | 0.519±0.009 | 0.515±0.008 | 0.518±0.007 | 0.507±0.008 |
> > > | | **ScalePU** | **0.598±0.008** | **0.569±0.028** | **0.560±0.049** | **0.521±0.018** |
> > > | CIFAR-10-1 | PU-AUC | 0.708±0.024 | 0.705±0.025 | 0.669±0.021 | 0.630±0.010 |
> > > | | **ScalePU** | **0.789±0.008** | **0.754±0.023** | **0.716±0.038** | **0.675±0.030** |
> > > | CIFAR-10-2 | PU-AUC | 0.686±0.070 | 0.672±0.058 | 0.567±0.047 | 0.522±0.084 |
> > > | | **ScalePU** | **0.777±0.020** | **0.719±0.068** | **0.712±0.048** | **0.593±0.034** |
> > > | F-MNIST-1 | PU-AUC | 0.626±0.024 | 0.694±0.037 | 0.682±0.036 | 0.707±0.026 |
> > > | | **ScalePU** | **0.929±0.005** | **0.901±0.022** | **0.865±0.016** | 0.775±0.039 |
> > > | F-MNIST-2 | PU-AUC | 0.690±0.049 | 0.750±0.085 | 0.698±0.042 | **0.675±0.086** |
> > > | | **ScalePU** | **0.907±0.017** | **0.857±0.037** | **0.742±0.045** | 0.583±0.050 |
> > > | STL-10-1 | PU-AUC | 0.660±0.021 | 0.655±0.039 | **0.618±0.036** | 0.587±0.035 |
> > > | | **ScalePU** | **0.713±0.038** | **0.684±0.042** | 0.616±0.062 | **0.603±0.054** |
> > > | STL-10-2 | PU-AUC | 0.685±0.048 | 0.631±0.037 | 0.567±0.019 | 0.583±0.033 |
> > > | | **ScalePU** | **0.705±0.038** | **0.647±0.060** | **0.611±0.042** | **0.584±0.043** |

---

### Official Review · Reviewer_7vEF · 2026-03-12

**Soundness:** 3
**Presentation:** 3
**Significance:** 4
**Originality:** 3
**Overall Recommendation:** 5
**Confidence:** 4

**Summary:**

This paper investigates the fundamental instability of PU learning in regimes where labeled positive examples are extremely scarce. To resolve this, the authors introduce ScalePU, a novel framework that incorporates variance regularization to induce a restricted hypothesis space and geometric regularization to foster representation compactness. The paper's core contribution is the theoretical derivation of the minimal sufficient learning threshold, which demonstrates how reducing Rademacher complexity and tightening the effective range enables stable estimation and achieves significant performance improvements under extreme scarcity of labeled positives.

**Compliance With Llm Reviewing Policy:**

Affirmed.

**Final Justification:**

My concerns are fully addressed so I'd like to keep my score.

**Key Questions For Authors:**

(1) When labeled positives are extremely scarce, a mini-batch might not contain any positive samples. It is unclear how $c_P$ is computed in such instances and whether repeatedly applying the variance penalty induces representation collapse.

(2) Could the authors provide a more detailed discussion on how a large class prior magnifies risk fluctuations when $n_P$ is minimal?

(3)The authors effectively shrink the hypothesis space $\mathcal{H}$ by reducing the variance and effective range, which successfully lowers the Rademacher complexity and the estimation error bound. However, from the perspective of standard statistical learning theory, overly restricting $\mathcal{H}$ typically increases the approximation bias. Could the authors provide a brief theoretical discussion on how the ScalePU balances this trade-off?

**Limitations:**

Yes

**Strengths And Weaknesses:**

Strengths:

(1) The authors rigorously characterize the instability of PU learning under label scarcity, and subsequently introduce the minimal sufficient learning threshold as a formal metric to quantify sample complexity requirements.

(2) The authors propose the ScalePU and establish solid theoretical guarantees through the derivation of generalization error bounds , formally demonstrating the efficacy of reducing Rademacher complexity and tightening the effective range.

(3) Beyond conventional risk estimation, the authors incorporate dual regularization to facilitate representation compactness in the feature space, with extensive experiments across eight diverse datasets validating the method's superior robustness under extreme scarcity of labeled positives.

Weaknesses:

(1) The appendix lacks the detailed mathematical derivation for Lemma 2.3.

(2) Experimental results suggest that under extreme label scarcity, the instability of risk estimation exhibits high sensitivity to the class prior $\pi$. The authors lack an analysis of how the prior specifically exacerbates the estimation variance.

(3) The proposed ScalePU framework relies primarily on the SCAR assumption, which limits its robustness in complex real-world scenarios where the labeled positive set exhibits selection bias.

---

> ### Author Rebuttal · Authors · 2026-03-30
>
> **Q1. Lack the detailed derivation for Lemma 2.3.**
>
> Thank you for your valuable comment. We will include the detailed proof for Lemma 2.3 in the revised appendix. The derivation relies on Dudley's entropy integral. For the original class $\mathcal{G} = \\{g_f: f \in \mathcal{H}\\}$, the complexity parameter $\kappa$ is governed by the integral of the covering number entropy over $[0, B]$. The variance constraint $\text{Var}\_P[g_f] \leq \tau$ restricts the hypothesis space to $\mathcal{G}\_\tau$, shrinking the effective $L_2(P)$ integration range from $[0, B]$ to $[0, \mathcal{O}(\sqrt{\tau})]$. Since $\mathcal{G}\_\tau \subseteq \mathcal{G}$, the covering number monotonically decreases, and combining this with the shrunken integration limit yields $\kappa_\tau \leq \kappa$ for sufficiently small $\tau$.
>
> &nbsp;
>
> **Q2. Explanation of how a large class prior $\pi$ magnifies risk fluctuations under extreme label scarcity.**
>
> Thank you for this insightful comment. In the standard PU formulation, the positive empirical term is explicitly scaled by $\pi$. As shown in Eq. (2), the uniform deviation bound is governed by $4\pi L\mathfrak{R}_{n_P}(\mathcal{H}) + \pi B\sqrt{\frac{2\ln(2/\delta)}{n_P}}$. **Because both terms scale linearly with $\pi$, the prior mathematically acts as a direct magnification factor.** For extremely small $n_P$, the $\mathcal{O}(1/\sqrt{n_P})$ factor blows up, and a large $\pi$ amplifies this estimation error, causing the unstable positive term to completely dominate the learning process. We note that **$\pi$ should remain within a practically reasonable range** since PU learning is fundamentally a binary classification problem. When $\pi$ approaches 0 or 1, the task degenerates toward detection rather than classification, and standard classification error ceases to be an appropriate metric. Our experimental settings reflect this, with $\pi$ ranging between 0.4 and 0.6 across all datasets.
>
> &nbsp;
>
> **Q3. Computation of $c_P$ in empty positive mini-batches and potential representation collapse from the variance penalty.**
>
> Thank you for your comment. Empty positive mini-batches do not occur in our framework because we **utilize two separate data loaders** that fetch from both positive and unlabeled sets simultaneously, which is standard practice in multi-dataset empirical risk minimization, where the objective is a sum of expectations over different distributions. Since $n_P$ is extremely small while $n_U$ is large, the two loaders rotate at very different speeds. At each iteration we draw one mini-batch from each independently; when the positive loader is exhausted it resets and cycles again, while epoch length is governed by the slower unlabeled loader. This asymmetric cycling guarantees every iteration receives a non-empty positive batch, making the computation of $c_P$ stable throughout training. To prevent representation collapse, **the variance penalty acts strictly on the functional outputs rather than directly compressing the feature space**, and the adaptive separation term explicitly prevents unlabeled representations from collapsing toward $c_P$.
>
> &nbsp;
>
> **Q4. Balancing the trade-off between restricting the hypothesis space and increasing approximation bias.**
>
> Thank you for this theoretical question. We respectfully suggest that the premise of this concern deserves closer examination. **The approximation error is determined solely by whether the optimal hypothesis in $\mathcal{H}$ is still contained in the restricted class.** If the restricted class retains the optimal solution, the approximation error is exactly unchanged, and the reduction in estimation error is obtained at no approximation cost. Our variance and geometric regularizations are constructed with reference to how well-performing members of $\mathcal{H}$ actually behave, so the restricted class is not an arbitrary subset but one designed to retain the region where good solutions reside. **The approximation-estimation tradeoff does not necessarily arise here**, because we are constructing a structured subset of a class that is already fixed.
>
> &nbsp;
>
> **Q5. Limitations of relying on the SCAR assumption.**
>
> Thank you for your comment. We acknowledge that ScalePU relies on the SCAR assumption, which is the standard foundation for representative methods such as uPU and nnPU. Our primary focus is addressing estimation instability under extreme label scarcity within this established paradigm. If the labeling mechanism is instance-dependent, the PU learning problem becomes mathematically unidentifiable without further structural assumptions. As Box's well-known principle reminds us, all models are wrong, but some are useful. SCAR is the simplest tractable assumption that renders the problem identifiable, and it has proven useful across a wide range of applications even when only approximately satisfied. Extending our framework to biased labeling scenarios represents a valuable direction for future work.

---

> > ### Author Rebuttal · Reviewer_7vEF · 2026-04-02
> >
> > My concerns are fully addressed so I'd like to keep my score.

---

### Official Review · Reviewer_jV2N · 2026-03-13

**Soundness:** 3
**Presentation:** 3
**Significance:** 3
**Originality:** 3
**Overall Recommendation:** 5
**Confidence:** 3

**Summary:**

This work introduces a Positive-Unlabeled learning (PU learning) method with variance regularization and geometric regularizations. With an argument analogous to Structural Risk Minimization (SRM), the authors argue that variance and geometric regularizations can reduce the number of labeled positives required for learning. Finally, through an extensive set of experiments, they show that their method outperforms state-of-the-art PU learning methods.

**Compliance With Llm Reviewing Policy:**

Affirmed.

**Key Questions For Authors:**

I think similar regularizations could also be applied in semi-supervised learning to reduce dependence on labeled data. I would be curious to see a comparison between the effect of these regularizations in PU learning and in standard semi-supervised learning. In addition, I would appreciate further discussion in the paper of semi-supervised learning more broadly, as well as existing methods aimed at reducing dependence on labeled data.

**Strengths And Weaknesses:**

In Theorems 2.4 and 2.5, the authors show—using ideas inspired by Structural Risk Minimization (SRM), that the minimum number of positive training examples required for learning can be reduced through the proposed regularizations. This analysis is interesting. However, I believe the true strength of this paper lies in its experimental evaluation. The experiments are conducted at a large scale and include comparisons with many existing PU learning algorithms, clearly demonstrating that ScalePU consistently outperforms these baselines. The presentation of the paper is also solid and generally easy to follow. Therefore, I recommend accepting this paper.

---

> ### Author Rebuttal · Authors · 2026-03-30
>
> **Response to Reviewer jV2N**
>
> &nbsp;&nbsp;&nbsp;&nbsp;Thank you for this insightful comment. We agree that this is a meaningful direction, and we appreciate the opportunity to discuss it.
>
> &nbsp;&nbsp;&nbsp;&nbsp;However, we would like to clarify that the specific forms of our regularizations in this paper are not general-purpose tools that happen to be applied here, but are derived from a PU-specific theoretical analysis. Recall that our key theoretical quantity is the minimal sufficient learning threshold, which depends on two structural parameters: the Rademacher complexity parameter $\kappa$ of the hypothesis class, and the uniform range bound $B$ of the positive term. The two regularizations we propose reduce $\kappa$ to $\kappa\tau$ and $B$ to $L_g\rho$ respectively, which jointly lower this threshold. The reason these two parameters arise, and the reason they are specifically associated with the positive term, is a direct consequence of the asymmetric structure of the PU risk estimator. The unlabeled term is well-estimated for any large $n_U$, while the positive term becomes the sole source of estimation instability as $n_P \to 0$. This asymmetry is what makes $\kappa$ and $B$ the correct targets for regularization in the PU setting.
>
> &nbsp;&nbsp;&nbsp;&nbsp;Moreover, there is a more fundamental structural difference between PU learning and SSL. Although PU learning involves unlabeled data, the unlabeled set in PU is not truly unlabeled in the SSL sense. Under the SCAR assumption, the unlabeled set is a mixture of positive and negative examples drawn from the marginal distribution, which means it functions as a noisy negative set. Contrastly, unlabeled data are genuinely unlabeled in SSL. This distinction is precisely what gives rise to the corrective term $g_f$ in the PU risk estimator. The function $g_f$ has no counterpart in SSL, because unlabeled data in SSL carry no such class-conditional contamination structure that requires correction.
>
> &nbsp;&nbsp;&nbsp;&nbsp;To make the contrast with SSL concrete, suppose one wished to derive analogous regularizations for SSL following the same two directions. In SSL, both the labeled positive risk $\hat{R}^+_L(f)$ and the labeled negative risk $\hat{R}^-_L(f)$ are present, and the generalization bound for the labeled term takes the symmetric form
>
> $$\sup_{f \in \mathcal{H}} \left|\hat{R}_L(f) - R_L(f)\right| \leq 4L\mathfrak{R}{n_L}(\mathcal{H}) + B\sqrt{\frac{2\ln(2/\delta)}{n_L}},$$
>
> where $n_L$ is the total number of labeled examples and $B$ is the uniform range of the full loss $\ell(yf(x))$ over both classes. Following the same logic as our paper, one could define a variance-based regularizer targeting $\kappa$ as
>
> $$\Omega^{\text{SSL}}\_{\text{var}}(f) = \widehat{\text{Var}}\_{L}\left[\ell(yf(x))\right] = \frac{1}{n_L}\sum_{i=1}^{n_L}\left(\ell(y_if(x_i)) - \bar{\ell}\right)^2,$$
>
> and a compactness regularizer targeting $B$ as
>
> $$\Omega^{\text{SSL}}_{\text{compact}}(h) = \frac{1}{n_L}\sum\_{i=1}^{n_L}\\|h(x_i) - c_L\\|^2,$$
>
> where $c_L$ is the centroid of all labeled representations. While these SSL regularizers share a superficial structural resemblance to ours, they are fundamentally different objects. In SSL, the labeled set contains examples from both classes, so the empirical risk is symmetric and both $\kappa$ and $B$ govern the generalization of the full joint loss $\ell(yf(x))$ equally across all labeled samples. There is no asymmetric instability to target, and no corrective quantity analogous to $g_f(x) = \ell(f(x)) - \ell(-f(x))$, because the labeled set in SSL always contains both classes by construction. The regularizers in SSL above would therefore reduce a standard symmetric generalization bound. It is a worthwhile goal, but one that bears no structural relationship to the problem our paper addresses.
>
> &nbsp;&nbsp;&nbsp;&nbsp;We are glad to add a discussion of this relationship to SSL in the related work or discussion section of the revised manuscript, making the above contrast explicit.
>
> &nbsp;&nbsp;&nbsp;&nbsp;Finally, to complement this discussion empirically, we have conducted a preliminary comparison between ScalePU and two representative SSL methods, MixMatch and FixMatch, in the low-label regime on six benchmark PU datasets. The results are available at the anonymous link (https://anonymous.4open.science/r/SSL-D89F/).

---

> > ### Author Rebuttal · Reviewer_jV2N · 2026-04-04
> >
> > I thank the authors for their detailed answer, and for extra experimentations they conducted. I will maintain my positive score.

---

### Decision · Program_Chairs · 2026-04-30

**Decision:**

Accept (regular)

**Comment:**

This paper studies PU learning when labeled positives are extremely scarce and identifies the positive risk term as the main source of instability via a Rademacher-style bound. It introduces the minimal sufficient learning threshold to quantify how many labeled positives are required for stable learning, and proposes ScalePU, which combines variance regularization (reducing effective Rademacher complexity) and geometric regularization (tightening the effective range). Experiments on eight benchmarks under low-positive regimes show consistent gains over strong PU baselines, with ablations confirming the contribution of each component.
All four reviewers recommend acceptance (5,5,5,4). They find the problem important, the method well-motivated and technically sound, and the empirical results strong. Concerns about missing derivations, the role of the class prior, SCAR, the interpretation/tightness of the threshold, the relation to SSL and PU-AUC methods, additional metrics (ROC-AUC), and computational overhead were addressed in a thorough rebuttal, with extra experiments (variance characterization, SSL comparison, ROC-AUC, PU-AUC, timing) and clearer discussion.

Remaining issues mainly concern the looseness of classical bounds and the gap between theoretical thresholds and practice, which are acknowledged as limitations. Overall, I recommend acceptance.